# Active Labeling: Streaming Stochastic Gradients

**Vivien Cabannes**[*]
Meta

**Francis Bach**
INRIA / ENS / PSL

**Vianney Perchet**
ENSAE

**Alessandro Rudi**
INRIA / ENS / PSL

## Abstract

The workhorse of machine learning is stochastic gradient descent. To access stochastic gradients, it is common to consider iteratively input/output pairs of a training dataset. Interestingly, it appears that one does not need full supervision to access stochastic gradients, which is the main motivation of this paper. After formalizing the "active labeling" problem, which focuses on active learning with partial supervision, we provide a streaming technique that provably minimizes the ratio of generalization error over the number of samples. We illustrate our technique in depth for robust regression.

## 1 Introduction

A large amount of the current hype around artificial intelligence was fueled by the recent successes of supervised learning. Supervised learning consists in designing an algorithm that maps inputs to outputs by learning from a set of input/output examples. When accessing many samples, and given enough computation power, this framework is able to tackle complex tasks. Interestingly, many of the difficulties arising in practice do not emerge from choosing the right statistical model to solve the supervised learning problem, but from the problem of collecting and cleaning enough data (see Chapters 1 and 2 of Géron, 2017, for example). Those difficulties are not disjoint from the current trends toward data privacy regulations (Council of European Union, 2016). This fact motivates this work, where we focus on how to efficiently collect information to carry out the learning process.

In this paper, we formalize the "active labeling" problem for weak supervision, where the goal is to learn a target function by acquiring the most informative dataset given a restricted budget for annotation. We focus explicitly on weak supervision that comes as a set of label candidates for each input, aiming to partially supervise input data in the most efficient way to guide a learning algorithm. We also restrict our study to the streaming variant where, for each input, only a single partial information can be collected about its corresponding output. The crux of this work is to leverage the fact that full supervision is not needed to acquire unbiased stochastic gradients, and perform stochastic gradient descent.

The following summarizes our contributions.

1. First, we introduce the "active labeling" problem, which is a relevant theoretical framework that encompasses many useful problems encountered by practitioners trying to annotate their data in the most efficient fashion, as well as its streaming variation, in order to deal with privacy preserving issues. This is the focus of Section 2.
2. Then, in Section 3, we give a high-level framework to access unbiased stochastic gradients with weak information only. This provides a simple solution to the streaming "active labeling" problem.
3. Finally, we detail this framework for a robust regression task in Section 4, and provide an algorithm whose optimality is proved in Section 5.

---

[*]Work done while at INRIA / ENS / PSL. Contact the first author at `vivien.cabannes@gmail.com`.

36th Conference on Neural Information Processing Systems (NeurIPS 2022).

As a proof of concept, we provide numerical simulations in Section 6. We conclude with a high-level discussion around our methods in Section 7.

**Related work.** Active query of information is relevant to many settings. The most straightforward applications are searching games, such as Bar Kokhba or twenty questions (Walsorth, 1882). We refer to Pelc (2002) for an in-depth survey of such games, especially when liars introduce uncertainty, and their relations with coding on noisy channels. But applications are much more diverse, *e.g.* for numerical simulation (Chevalier et al., 2014), database search (Qarabaqi and Riedewald, 2014), or shape recognition (Geman and Jedynak, 1993), to name a few.

In terms of motivations, many streams of research can be related to this problem, such as experimental design (Chernoff, 1959), statistical queries (Kearns, 1998; Fotakis et al., 2021), crowdsourcing (Doan et al., 2011), or aggregation methods in weak supervision (Ratner et al., 2020). More precisely, "active labeling"[2] consists in having several inputs and querying partial information on the labels. It is close to active learning (Settles, 2010; Dasgupta, 2011; Hanneke, 2014), where there are several inputs, but exact outputs are queried; and to active ranking (Valiant, 1975; Ailon, 2011; Braverman et al., 2019), where partial information is queried, but there is only one input. The streaming variant introduces privacy preserving constraints, a problem that is usually tackled through the notion of differential privacy (Dwork et al., 2006).

In terms of formalization, we build on the partial supervision formalization of Cabannes et al. (2020), which casts weak supervision as sets of label candidates and generalizes semi-supervised learning (Chapelle et al., 2006). Finally, our sequential setting with a unique final reward is similar to combinatorial bandits in a pure-exploration setting (Garivier and Kaufmann, 2016; Fiez et al., 2019).

## 2 The "active labeling" problem

Supervised learning is traditionally modeled in the following manner. Consider $\mathcal{X}$ an input space, $\mathcal{Y}$ an output space, $\ell : \mathcal{Y} \times \mathcal{Y} \to \mathbb{R}$ a loss function, and $\rho \in \Delta_{\mathcal{X} \times \mathcal{Y}}$ a joint probability distribution. The goal is to recover the function

$$f^* \in \underset{f:\mathcal{X} \to \mathcal{Y}}{\arg\min} \mathcal{R}(f) := \mathbb{E}_{(X,Y) \sim \rho}[\ell(f(X), Y)], \tag{1}$$

yet, without accessing $\rho$, but a dataset of independent samples distributed according to $\rho$, $\mathcal{D}_n = (X_i, Y_i)_{i \le n} \sim \rho^{\otimes n}$. In practice, accessing data comes at a cost, and it is valuable to understand the cheapest way to collect a dataset allowing to discriminate $f^*$.

We shall suppose that the input data $(X_i)_{i \le n}$ are easy to collect, yet that labeling those inputs to get outputs $(Y_i)_{i \le n}$ demands a high amount of work. For example, it is relatively easy to scrap the web or medical databases to access radiography images, but labeling them by asking radiologists to recognize tumors on zillions of radiographs will be both time-consuming and expensive. As a consequence, *we assume the $(X_i)_{i \le n}$ given but the $(Y_i)_{i \le n}$ unknown.* As getting information on the labels comes at a cost (*e.g.*, paying a pool of label workers, or spending your own time), given a budget constraint, what information should we query on the labels?

To quantify this problem, we will assume that *we can sequentially and adaptively query $T$ information of the type $\mathbf{1}_{Y_{i_t} \in S_t}$, for any index $i_t \in \{1, \cdots, n\}$ and any set of labels $S_t \subset \mathcal{Y}$ (belonging to a specified set of subsets of $\mathcal{Y}$).* Here, $t \in \{1, \cdots, T\}$ indexes the query sequence, and $T \in \mathbb{N}$ is a fixed budget. The goal is to optimize the design of the sequence $(i_t, S_t)$ in order to get the best estimate of $f^*$ in terms of risk minimization (1). In the following, we give some examples to make this setting more concrete.

**Example 1** (Classification with attributes). *Suppose that a labeler is asked to provide fine-grained classes on images (Krause et al., 2016; Zheng et al., 2019), such as the label "caracal" in Figure 6. This would be difficult for many people. Yet, it is relatively easy to recognize that the image depicts a "feline" with "tufted-ears" and "sandy color". As such, a labeler can give the weak information that Y belongs to the set "feline", $S_1 = \{$"cat", "lion", "tiger", ...$\}$, and the set "tufted ears", $S_2 = \{$"Great horned owl", "Aruacana chicken", ...$\}$. This is enough to recognize that $Y \in S_1 \cap S_2 = \{$"caracal"$\}$. The question $\mathbf{1}_{Y \in S_1}$, corresponds to asking if the image depicts a feline.*

---

[2]Note that the wording "active labeling" has been more or less used as synonymous of "active learning" (*e.g.*, Wang and Shang, 2014). In contrast, we use "active labeling" to design "active weakly supervised learning".

*Literature on hierarchical classification and autonomic taxonomy construction provides interesting ideas for this problem* (e.g., *Cesa-Bianchi et al., 2006; Gangaputra and Geman, 2006*).

**Example 2** (Ranking with partial ordering)**.** *Consider a problem where for a given input x, characterizing a user, we are asked to deduce their preferences over m items. Collecting such a label requires knowing the exact ordering of the m items induced by a user. This might be hard to ask for. Instead, one can easily ask the user which items they prefer in a collection of a few items. The user's answer will give weak information about the labels, which can be modeled as knowing $\mathbf{1}_{Y_i \in S} = 1$, for S the set of total orderings that satisfy this partial ordering. We refer the curious reader to active ranking and dueling bandits for additional contents (Jamieson and Nowak, 2011; Bengs et al., 2021).*

**Example 3** (Pricing a product)**.** *Suppose that we want to sell a product to a consumer characterized by some features x, this consumer is ready to pay a price $y \in \mathbb{R}$ for this product. We price it $f(x) \in \mathbb{R}$, and we observe $\mathbf{1}_{f(x) < y}$, that is if the consumer is willing to buy this product at this price tag or not (Cesa-Bianchi et al., 2019; Liu et al., 2021). Although, in this setting, the goal is often to minimize the regret, which contrasts with our pure exploration setting.*

As a counter-example, our assumptions are not set to deal with missing data, *i.e.* if some coordinates of some input feature vectors $X_i$ are missing (Rubin, 1976). Typically, this happens when input data comes from different sources (*e.g.*, when trying to predict economic growth from country information that is self-reported).

**Streaming variation.** The special case of the active labeling problem we shall consider consists in its variant without resampling. This corresponds to the online setting where one can only ask one question by sample, formally $i_t = t$. This setting is particularly appealing for privacy concerns, in settings where the labels $(Y_i)$ contain sensitive information that should not be revealed totally. For example, some people might be more comfortable giving a range over a salary rather than the exact value; or in the context of polling, one might not call back a previous respondent characterized by some features $X_i$ to ask them again about their preferences captured by $Y_i$. Similarly, the streaming setting is relevant for web marketing, where inputs model new users visiting a website, queries model sets of advertisements chosen by an advertising company, and one observes potential clicks.

## 3 Weak information as stochastic gradients

In this section, we discuss how unbiased stochastic gradients can be accessed through weak information.

Suppose that we model $f = f_\theta$ for some Hilbert space $\Theta \ni \theta$. With some abuse of notations, let us denote $\ell(x, y, \theta) := \ell(f_\theta(x), y)$. We aim to minimize $\mathcal{R}(\theta) = \mathbb{E}_{(X,Y)}[\ell(X, Y, \theta)]$. Assume that $\mathcal{R}$ is differentiable (or sub-differentiable) and denote its gradients by $\nabla_\theta \mathcal{R}$.

**Definition 1** (Stochastic gradient)**.** *A stochastic gradient of $\mathcal{R}$ is any random function $G : \Theta \to \Theta$ such that $\mathbb{E}[G(\theta)] = \nabla_\theta \mathcal{R}(\theta)$. Given some step size function $\gamma : \mathbb{N} \to \mathbb{R}^*$, a stochastic gradient descent (SGD) is a procedure, $(\theta_t) \in \Theta^\mathbb{N}$, initialized with some $\theta_0$ and updated as $\theta_{t+1} = \theta_t - \gamma(t)G(\theta_t)$, where the realization of $G(\theta_t)$ given $\theta_t$ is independent of the previous realizations of $G(\theta_s)$ given $\theta_s$.*

In supervised learning, SGD is usually performed with the stochastic gradients $\nabla_\theta \ell(X, Y, \theta)$. More generally, stochastic gradients are given by

$$G(\theta) = \mathbf{1}_{\nabla_\theta \ell(X,Y,\theta) \in T} \cdot \tau(T), \tag{2}$$

for $\tau : \mathcal{T} \to \Theta$ with $\mathcal{T} \subset 2^\Theta$ a set of subsets of $\Theta$, and $T$ a random variable on $\mathcal{T}$, such that

$$\forall \theta \in \Theta, \quad \mathbb{E}_T[\mathbf{1}_{\theta \in T} \cdot \tau(T)] = \theta. \tag{3}$$

Stated otherwise, if you have a way to image a vector $\theta$ from partial measurements $\mathbf{1}_{\theta \in T}$ such that you can reconstruct this vector in a linear fashion (3), then it provides you a generic strategy to get an unbiased stochastic estimate of this vector from a partial measurement (2).

For $\psi : \mathcal{Y} \to \Theta$ a function from $\mathcal{Y}$ to $\Theta$ (*e.g.*, $\psi = \nabla_\theta(X, \cdot, \theta)$), a question $\mathbf{1}_{\psi(Y) \in T}$ translates into a question $\mathbf{1}_{Y \in S}$ for some set $S = \psi^{-1}(T) \subset \mathcal{Y}$, meaning that the stochastic gradient (2) can be evaluated from a single query. As a proof of concept, we derive a generic implementation for $T$ and $\tau$ in Appendix B. This provides a generic SGD scheme to learn functions from weak queries when there are no constraints on the sets to query.

**Remark 2** (Cutting plane methods). *While we provide here a descent method, one could also develop cutting-plane/ellipsoid methods to localize $\theta^*$ according to weak information, which corresponds to the techniques developed for pricing by Cohen et al. (2020) and related literature.*

## 4 Median regression

In this section, we focus on efficiently acquiring weak information providing stochastic gradients for regression problems. In particular, we motivate and detail our methods for the absolute deviation loss.

Motivated by seminal works on censored data (Tobin, 1958), we shall suppose that *we query half-spaces*. For an output $y \in \mathcal{Y} = \mathbb{R}^m$, and any hyper-plane $z + u^\perp \subset \mathbb{R}^m$ for $z \in \mathbb{R}^m$, $u \in \mathbb{S}^{m-1}$, we can ask a labeler to tell us which half-space $y$ belongs to. Formally, *we access the quantity* $\text{sign}(\langle y - z, u \rangle)$ *for a given unit cost*. Such an imaging scheme where one observes summations of its components rather than a vector itself bears similarity with compressed sensing. To provide further illustration, this setting could help to price products while selling bundles: where the context $x$ characterizes some users, web-pages or/and advertisement companies; the label $y \in \mathbb{R}^m$ corresponds to the value associated to $m$ different items, such as stocks composing an index, or advertisement spots; and the observation $\text{sign}(\langle y, u \rangle - c)$ (with $c = \langle z, u \rangle$) captures if the user $x$ buys the basket with weights $u \in \mathbb{S}^{m-1}$ when it is priced $c$.

**Least-squares.** For regression problems, it is common to look at the mean square loss

$$\ell(X, Y, \theta) = \|f_\theta(X) - Y\|^2, \qquad \nabla_\theta \ell(X, Y, \theta) = 2(f_\theta(X) - Y)^\top D f_\theta(X),$$

where $D f_\theta(x) \in \mathcal{Y} \otimes \Theta$ denotes the Jacobian of $\theta \to f_\theta(x)$. In rich parametric models, it is preferable to ask questions on $Y \in \mathcal{Y}$ rather than on gradients in $\Theta$ which is a potentially much bigger space. If we assume that $Y$ and $f_\theta(X)$ are bounded in $\ell^2$-norm by $M \in \mathbb{R}_+$, we can adapt (2) and (3) through the fact that for any $z \in \mathcal{Y}$, such that $\|z\| \leq 2M$, as proven in Appendix B,

$$\mathbb{E}_{U,V} \left[ \mathbf{1}_{\langle z, U \rangle \geq V} \cdot U \right] = c_1 \cdot z, \quad \text{where} \quad c_1 = \mathbb{E}_{U,V} \left[ \mathbf{1}_{\langle e_1, U \rangle \geq V} \cdot \langle e_1, U \rangle \right] = \frac{\pi^{3/2}}{2M(m^2 + 4m + 3)},$$

for $U$ uniform on the sphere $\mathbb{S}^{m-1}$ and $V$ uniform on $[0, 2M]$. Applied to $z = f_\theta(X) - Y$, it designs an SGD procedure by querying information of the type $\mathbf{1}_{\langle Y, U \rangle < \langle f_\theta(X), U \rangle - V}$.

**A case for median regression.** Motivated by robustness purposes, we will rather expand on median regression. In general, we would like to learn a function that, given an input, replicates the output of I/O samples generated by the joint probability $\rho$. In many instances, $X$ does not characterize all the sources of variations of $Y$, *i.e.* input features are not rich enough to characterize a unique output, leading to randomness in the conditional distributions $(Y|X)$. When many targets can be linked to a vector $x \in \mathcal{X}$, how to define a consensual $f(x)$? For analytical reasons, statisticians tend to use the least-squares error which corresponds to asking for $f(x)$ to be the mean of the distribution $(Y|X = x)$. Yet, means are known to be too sensitive to rare but large outputs (see *e.g.*, Huber, 1981), and cannot be defined as good and robust consensus in a world of heavy-tailed distributions. This contrasts with the median, which, as a consequence, is often much more valuable to summarize a range of values. For instance, median income is preferred over mean income as a population indicator (see *e.g.*, US Census Bureau, 2021).

**Median regression.** The geometric median is variationally defined through the absolute deviation loss, leading to

$$\ell(X, Y, \theta) = \|f_\theta(X) - Y\|, \qquad \nabla_\theta \ell(X, Y, \theta) = \left( \frac{f_\theta(X) - Y}{\|f_\theta(X) - Y\|} \right)^\top D f_\theta(X). \tag{4}$$

Similarly to the least-squares case, we can access weakly supervised stochastic gradients through the fact that for $z \in \mathbb{S}^{m-1}$, as shown in Appendix B,

$$\mathbb{E}_U \left[ \text{sign}(\langle z, U \rangle) \cdot U \right] = c_2 \cdot z, \quad \text{where} \quad c_2 = \mathbb{E}_U \left[ \text{sign}(\langle e_1, U \rangle) \cdot \langle e_1, U \rangle \right] = \frac{\sqrt{\pi} \Gamma(\frac{m-1}{2})}{m \Gamma(\frac{m}{2})}, \tag{5}$$

where $U$ is uniformly drawn on the sphere $\mathbb{S}^{m-1}$, and $\Gamma$ is the gamma function. This suggests Algorithm 1.

---
**Algorithm 1:** Median regression with SGD.

---
**Data:** A model $f_\theta$ for $\theta \in \Theta$, some data $(X_i)_{i \leq n}$, a labeling budget $T$, a step size rule $\gamma : \mathbb{N} \to \mathbb{R}_+$
**Result:** A learned parameter $\hat\theta$ and the predictive function $\hat{f} = f_{\hat\theta}$.
Initialize $\theta_0$.
**for** $t \leftarrow 1$ **to** $T$ **do**
    Sample $U_t$ uniformly on $\mathbb{S}^{m-1}$.
    Query $\varepsilon = \text{sign}(\langle Y_t - z, U_t\rangle)$ for $z = f_{\theta_{t-1}}(X_t)$.
    Update the parameter $\theta_t = \theta_{t-1} + \gamma(t)\varepsilon \cdot U_t^\top (Df_{\theta_{t-1}}(X_t))$.
Output $\hat\theta = \theta_T$, or some average, *e.g.*, $\hat\theta = T^{-1} \sum_{t=1}^T \theta_t$.

---

# 5  Statistical analysis

In this section, we quantify the performance of Algorithm 1 by proving optimal rates of convergence when the median regression problem is approached with (reproducing) kernels. For simplicity, we will assume that $f^*$ can be parametrized by a linear model (potentially of infinite dimension).

**Assumption 1.** *Assume that the solution $f^* : \mathcal{X} \to \mathbb{R}^m$ of the median regression problem* (1) *and* (4) *can be parametrized by some separable Hilbert space $\mathcal{H}$, and a bounded feature map $\varphi : \mathcal{X} \to \mathcal{H}$, such that, for any $i \in [m]$, there exists some $\theta_i^* \in \mathcal{H}$ such that $\langle f^*(\cdot), e_i\rangle_{\mathcal{Y}} = \langle \theta_i^*, \varphi(\cdot)\rangle_{\mathcal{H}}$, where $(e_i)$ is the canonical basis of $\mathbb{R}^m$. Written into matrix form, there exists $\theta^* \in \mathcal{Y} \otimes \mathcal{H}$, such that $f^*(\cdot) = \theta^*\varphi(\cdot)$.*

The curious reader can easily relax this assumption in the realm of reproducing kernel Hilbert spaces following the work of Pillaud-Vivien et al. (2018a). Under the linear model of Assumption 1, Algorithm 1 is specified with $u^\top Df_\theta(x) = u \otimes \varphi(x)$. Note that rather than working with $\Theta = \mathcal{Y} \otimes \mathcal{H}$ which is potentially infinite-dimensional, empirical estimates can be represented in the finite-dimensional space $\mathcal{Y} \otimes \text{Span}\{\varphi(X_i)\}_{i \leq n}$, and well approximated by small-dimensional spaces to ensure efficient computations (Williams and Seeger, 2000; Meanti et al., 2020).

One of the key points of SGD is that gradient descent is so gradual that one can use noisy or stochastic gradients without loosing statistical guarantees while speeding up computations. This is especially true when minimizing convex functions that are nor strongly-convex, *i.e.*, bounded below by a quadratic, nor smooth, *i.e.*, with Lipschitz-continuous gradient (see, *e.g.*, Bubeck, 2015). In particular, the following theorem, proven in Appendix A.1, states that Algorithm 1 minimizes the population risk at a speed at least proportional to $O(T^{-1/2})$.

**Theorem 1** (Convergence rates). *Under Assumption 1, and under the knowledge of $\kappa$ and $M$ two real values such that $\mathbb{E}[\|\varphi(X)\|^2] \leq \kappa^2$ and $\|\theta^*\| \leq M$, with a budget $T \in \mathbb{N}$, a constant step size $\gamma = \frac{M}{\kappa\sqrt{T}}$ and the average estimate $\hat\theta = \frac{1}{T}\sum_{t=0}^{T-1}\theta_t$, Algorithm 1 leads to an estimate $f$ that suffers from an excess of risk*

$$\mathbb{E}\left[\mathcal{R}\left(f_{\hat\theta}\right)\right] - \mathcal{R}(f^*) \leq \frac{2\kappa M}{c_2\sqrt{T}} \leq \kappa M m^{3/2} T^{-1/2}, \tag{6}$$

*where the expectation is taken with respect to the randomness of $\hat\theta$ that depends on the dataset $(X_i, Y_i)$ as well as the questions $(i_t, S_t)_{t \leq T}$.*

While we give here a result for a fixed step size, one could retake the extensive literature on SGD to prove similar results for decaying step sizes that do not require to know the labeling budget in advance (*e.g.* setting $\gamma(t) \propto t^{-1/2}$ at the expense of an extra term in $\log(T)$ in front of the rates), as well as different averaging strategies (see *e.g.*, Bach, 2023). In practice, one might not know *a priori* the parameter $M$ but could nonetheless find the right scaling for $\gamma$ based on cross-validation.

The rate in $O(T^{-1/2})$ applies more broadly to all the strategies described in Section 3 as long as the loss $\ell$ and the parametric model $f_\theta$ ensure that $\mathcal{R}(\theta)$ is convex and Lipschitz-continuous. Although the constants appearing in front of rates depend on the complexity to reconstruct the full gradient $\nabla_\theta \ell(f_\theta(X_i, Y_i))$ from the reconstruction scheme (3). Those constants correspond to the second moment of the stochastic gradient. For example, for the least-squares technique described earlier one would have to replace $c_2$ by $c_1$ in (6).

Theorem 2, proven in Appendix A.3, states that any algorithm that accesses a fully supervised learning dataset of size $T$ cannot beat the rates in $O(T^{-1/2})$, hence any algorithm that collects weaker information on $(Y_i)_{i \leq T}$ cannot display better rates than the ones verified by Algorithm 1. This proves minimax optimality of our algorithm up to constants.

**Theorem 2** (Minimax optimality). *Under Assumption 1 and the knowledge of an upper bound on $\|\theta^*\| \leq M$, assuming that $\varphi$ is bounded by $\kappa$, there exists a universal constant $c_3$ such that for any algorithm $\mathcal{A}$ that takes as input $\mathcal{D}_T = (X_i, Y_i)_{i \leq T} \sim \rho^{\otimes T}$ for any $T \in \mathbb{N}$ and output a parameter $\theta$,*

$$\sup_{\rho \in \mathcal{M}_M} \mathbb{E}_{\mathcal{D}_T \sim \rho^{\otimes T}} \left[ \mathcal{R}(f_{\mathcal{A}(\mathcal{D}_T; \rho)}) \right] - \mathcal{R}(f_\rho; \rho) \geq c_3 M \kappa T^{-1/2}. \tag{7}$$

*The supremum over $\rho \in \mathcal{M}_M$ has to be understood as the supremum over all distributions $\rho \in \Delta_{\mathcal{X} \times \mathcal{Y}}$ such that the problem defined through the risk $\mathcal{R}(f; \rho) := \mathbb{E}_\rho[\ell(f(X), Y)]$ is minimized for $f_\rho$ that verifies Assumption 1 with $\|\theta^*\|$ bounded by a constant $M$.*

The same theorem applies for least-squares with a different universal constant. It should be noted that minimax lower bounds are in essence quantifying worst cases of a given class of problems. In particular, to prove Theorem 2, we consider distributions that lead to hard problems; more specifically, we assumed the variance of the conditional distribution $(Y \mid X)$ to be high. The practitioner should keep in mind that it is possible to add additional structure on the solution, leverage active learning or semi-supervised strategy such as uncertainty sampling (Nguyen et al., 2021), or Laplacian regularization (Zhu et al., 2003; Cabannes et al., 2021a), and reduce the optimal rates of convergence.

To conclude this section, let us remark that most of our derivations could easily be refined for practitioners facing a slightly different cost model for annotation. In particular, they might prefer to perform batches of annotations before updating $\theta$ rather than modifying the question strategy after each input annotation. This would be similar to mini-batching in gradient descent. Indeed, the dependency of our result on the annotation cost model and on Assumption 1 should not be seen as a limitation but rather as a proof of concept.

## 6 Numerical analysis

In this section, we illustrate the differences between our active method versus a classical passive method, for regression and classification problems. Further discussions are provided in Appendix E. Our code is available online at `https://github.com/VivienCabannes/active-labeling`.

Let us begin with the regression problem that consists in estimating the function $f^*$ that maps $x \in [0, 1]$ to $\sin(2\pi x) \in \mathbb{R}$. Such a regular function, which belongs to any Hölder or Sobolev classes of functions, can be estimated with the Gaussian kernel, which would ensure Assumption 1, and that corresponds to a feature map $\varphi$ such that $k(x, x') := \langle \varphi(x), \varphi(x') \rangle = \exp(- |x - x'| / (2\sigma^2))$ for any bandwidth parameter $\sigma > 0$.[3] On Figure 1, we focus on estimating $f^*$ given data $(X_i)_{i \in [T]}$ that are uniform on $[0, 1]$ in the noiseless setting where $Y_i = f^*(X_i)$, based on the minimization of the absolute deviation loss. The passive baseline consists in randomly choosing a threshold $U_i \sim \mathcal{N}(0, 1)$ and acquiring the observations $(\mathbf{1}_{Y_i > U_i})_{i \in [T]}$ that can be cast as the observation of the half-space $S_i = \left\{ y \in \mathcal{Y} \mid \mathbf{1}_{y > U_i} = \mathbf{1}_{Y_i > U_i} \right\} =: s(Y_i, U_i)$. In this noiseless setting, a good baseline to learn $f^*$ from the data $(X_i, S_i)$ is provided by the infimum loss characterization (see Cabannes et al., 2020)

$$f^* = \arg \min_{f: \mathcal{X} \to \mathcal{Y}} \mathbb{E}_{(X, S)} \left[ \inf_{y \in S} \ell(f(X), y) \right],$$

where the distribution over $X$ corresponds to the marginal of $\rho$ over $\mathcal{X}$, and the distribution over $(S \mid X = x)$ is the pushforward of $U \sim \mathcal{N}(0, 1)$ under $s(f^*(x), \cdot)$. The left plot on Figure 1 corresponds to an instance of SGD on such an objective based on the data $(X_i, S_i)$, while the right plot corresponds to Algorithm 1. We take the same hyperparameters for both plots, a bandwidth $\sigma = 0.2$ and an SGD step size $\gamma = 0.3$. We refer the curious reader to Figure 7 in Appendix E for plots illustrating the streaming history, and to Figure 10 for "real-world" experiments.

To illustrate the versatility of our method, we approach a classification problem through the median surrogate technique presented in Proposition 3. To do so, we consider the classification problem

---

[3]A noteworthy computational aspect of linear models, often refer as the "kernel trick", is that the features map $\varphi$ does not need to be explicit, the knowledge of $k : \mathcal{X} \times \mathcal{X} \to \mathbb{R}$ being sufficient to compute all quantities of interest (Scholkopf and Smola, 2001). This "trick" can be applied to our algorithms.

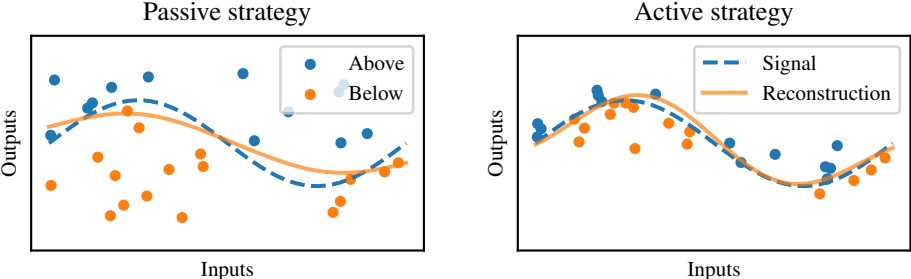

**Figure 1:** *Visual comparison of active and passive strategies.* Estimation in orange of the original signal $f^*$ in dashed blue based on median regression in a noiseless setting. Any orange point $(x, u) \in \mathbb{R}^2$ corresponds to an observation made that $u$ is below $f^*(x)$, while a blue point corresponds to $u$ above $f^*(x)$. The passive strategy corresponds to acquiring information based on $(U \mid x)$ following a normal distribution, while the active strategy corresponds to $(u \mid x) = f_\theta(x)$. The active strategy reconstructs the signal much better given the budget of $T = 30$ observations.

with $m \in \mathbb{N}$ classes, $\mathcal{X} = [0, 1]$ and the conditional distribution $(Y \mid X)$ linearly interpolating between Dirac in $y_1$, $y_2$ and $y_3$ respectively for $x = 0$, $x = 1/2$ and $x = 1$ and the uniform distribution for $x = 1/4$ and $x = 3/4$; and $X$ uniform on $\mathcal{X} \setminus ([1/4 - \varepsilon, 1/4 + \varepsilon] \cup [3/4 - \varepsilon, 3/4 + \varepsilon])$.

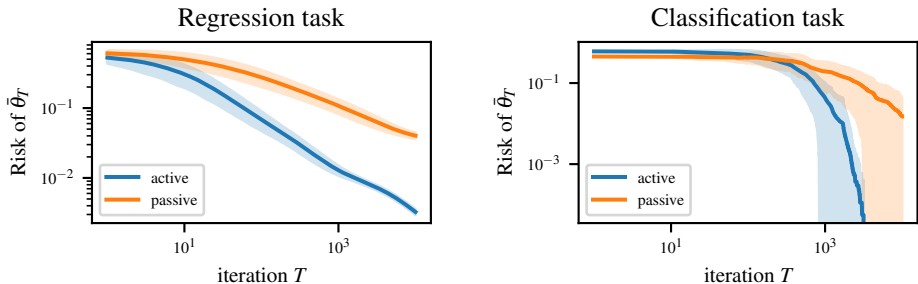

**Figure 2:** *Comparison of generalization errors of passive and active strategies* as a function of the annotation budget $T$. This error is computed by averaging over 100 trials. In solid is represented the average error, while the height of the dark area represents one standard deviation on each side. In order to consider the streaming setting where $T$ is not known in advance, we consider the decreasing step size $\gamma(t) = \gamma_0/\sqrt{t}$; and to smooth out the stochasticity due to random gradients, we consider the average estimate $\bar{\theta}_t = (\theta_1 + \cdots + \theta_t)/t$. The left figure corresponds to the noiseless regression setting of Figure 1, with $\gamma_0 = 1$. We observe the convergence behavior in $O(T^{-1/2})$ of our active strategy. The right setting corresponds to the classification problem setting described in the main text with $m = 100$, $\varepsilon = 1/20$, and approached with the median surrogate. We observe the exponential convergence phenomenon described by Pillaud-Vivien et al. (2018b); Cabannes et al. (2021b); its kicks in earlier for the active strategy. The two plots are displayed with logarithmic scales on both axes.

# 7 Discussion

## 7.1 Discrete output problems

In this section, we discuss casting Algorithm 1 into a procedure to tackle discrete-output problems, by leveraging surrogate regression tasks.

Learning problems with discrete output spaces are not as well understood as regression problems. This is a consequence of the complexity of dealing with combinatorial structures in contrast with continuous metric spaces. In particular, gradients are not defined for discrete output models. The current state-of-the-art framework to deal with discrete output problems is to introduce a continuous surrogate problem whose solution can be decoded as a solution on the original problem (Bartlett et al.,

2006). For example, one could solve a classification task with a median regression surrogate problem, which is the object of the next proposition, proven in Appendix C.

**Proposition 3** (Consistency of median surrogate). *The classification setting where $\mathcal{Y}$ is a finite space, and $\ell : \mathcal{Y} \times \mathcal{Y} \to \mathbb{R}$ is the zero-one loss $\ell(y, z) = \mathbf{1}_{y \neq z}$ can be solved as a regression task through the simplex embedding of $\mathcal{Y}$ in $\mathbb{R}^{\mathcal{Y}}$ with the orthonormal basis $(e_y)_{y \in \mathcal{Y}}$. More precisely, if $g^* : \mathcal{X} \to \mathbb{R}^{\mathcal{Y}}$ is the minimizer of the median surrogate risk $\mathcal{R}_S(g) = \mathbb{E}\left[\|g(X) - e_Y\|\right]$, then $f^* : \mathcal{X} \to \mathcal{Y}$ defined as $f^*(x) = \arg\max_{y \in \mathcal{Y}} g_y^*(x)$ minimizes the original risk $\mathcal{R}(f) = \mathbb{E}\left[\ell(f(X), Y)\right]$.* [4]

More generally, any discrete output problem can be solved by reusing the consistent least-squares surrogate of Ciliberto et al. (2020). Algorithm 1 can be adapted to the least-squares problem based on specifications at the beginning of Section 4. This allows using our method in an off-the-shelve fashion for all discrete output problems. For example, a problem consisting in ranking preferences over $m$ items can be approached with the Kendall correlation loss $\ell(y, z) = -\varphi(y)^\top \varphi(z)$ with $\varphi(y) = (1_{y(i) > y(j)})$ for $i < j \leq m$, where $y$ and $z$ are permutations over $[m]$ that encode the rank of each element in terms of user preferences. In this setting, the surrogate task introduced by Ciliberto et al. (2020) consists in learning $g(x) = \mathbb{E}[\varphi(Y)|X = x]$ as a least-squares problem. The half-space surrogate queries translate directly into the questions $\sum_{i < j \leq m} w(i, j) 1_{y(i) > y(j)} > c$ for some $(w(i, j)), c$ in $\mathbb{R}$. In particular, if $U$ is chosen to be uniform on the canonical basis (rather than on the sphere), those questions translate into pairwise orderings (*e.g.*, does user $x$ prefer movie $i$ or movie $j$?). In terms of guarantee akin to Theorem 1, retaking the calibration inequality of Ciliberto et al. (2020), we get convergence rates of the form $m^{3/2} T^{-1/4}$. In terms of guarantee akin to Theorem 2, since we need as least $\log_2(m!) \simeq m \log(m)$ binary queries to discriminate between $m!$ permutations, we can expect a lower bound in $m^{1/2} \log(m)^{1/2} T^{-1/2}$. More generally, many ranking problems can be approached with correlation losses and tackled through surrogate regression problems on the convex hulls of some well-known polytopes such as the Birkhoff polytope or the permutohedron (*e.g.*, Ailon, 2014). Although their descriptions is out-of-scope of this paper, linear cuts of those polytopes form well-structured queries sets – *e.g.*, the faces of all dimensions of the permutohedron correspond, in a one-to-one fashion, to strict weak orderings (Ziegler, 1995).

In those discrete settings, Theorem 1 can be refined under low noise conditions. In particular, under generalization of the Massart noise condition, our approach could even exhibit exponential convergence rates as illustrated on Figure 2. For classification problems, this condition can be expressed as the existence of a threshold $\delta > 0$ such that for almost all $x \in \mathcal{X}$ and $z \in \mathcal{Y}$, we have $\mathbb{P}(Y = f(x)|X = x) - \mathbb{P}(Y = z|X = x) \notin (0, \delta)$. Arguably, this assumption is met on well-curated images dataset such as ImageNet or CIFAR10, where for each input $X$ the most probable class has always more than *e.g.* 60% of chance to be the target $Y$. When this assumption holds together with Assumption 1 (when the surrogate target $g^*$ belongs to the RKHS and the kernel is bounded), then the right hand-side of equation (6) can be replaced by $\exp(-cT)$ for some constant $c$. The proof would be a simple adaptation of Pillaud-Vivien et al. (2018a); Cabannes et al. (2021b) to our case.

## 7.2 Supervised learning baseline with resampling

In this section, we discuss simple supervised learning baselines that compete with Algorithm 1 when resampling is allowed.

When resampling is allowed a simple baseline for the active labeling problem is provided by supervised learning. In regression problems with the query of any half-space, a method that consists in annotating each $(Y_i)_{i \leq n(T, \varepsilon)}$ up to precision $\varepsilon$, before using any supervised learning method to learn $f$ from $(X_i, Y_i)_{i \leq n(T, \varepsilon)}$ could acquire $n(T, \varepsilon) \simeq T/m \log_2(\varepsilon^{-1})$ data points with a dichotomic search along all directions, assuming $Y_i$ bounded or sub-Gaussian. In terms of minimax rates, such a procedure cannot perform better than in $n(T, \varepsilon)^{-1/2} + \varepsilon$, the first term being due to the statistical limit in Theorem 2, the second due to the incertitude $\varepsilon$ on each $Y_i$ that transfers to the same level of incertitude on $f$. Optimizing with respect to $\varepsilon$ yields a bound in $O(T^{-1/2} \log(T)^{1/2})$. Therefore, this not-so-naive baseline is only suboptimal by a factor $\log(T)^{1/2}$. In the meanwhile, Algorithm 1 can be rewritten with resampling, as well as Theorem 1, which we prove in Appendix A.2. Hence, our technique will still achieve minimax optimality for the problem "with resampling". In other terms, by deciding

---

[4] As a side note, while we are not aware of any generic theory encompassing the absolute-deviation surrogate of Proposition 3, we showcase its superiority over least-squares on at least two types of problems on Figures 3 and 4 in Appendix C.

to acquire more imprecise information, our algorithm reduces annotation cost for a given level of generalization error (or equivalently reduces generalization error for a given annotation budget) by a factor $\log(T)^{1/2}$ when compared to this baseline.

The picture is slightly different for discrete-output problems. If one can ask any question $s \in 2^{\mathcal{Y}}$ then with a dichotomic search, one can retrieve any label with $\log_2(m)$ questions. Hence, to theoretically beat the fully supervised baseline with the SGD method described in Section 3, one would have to derive a gradient strategy (2) with a small enough second moment (*e.g.*, for convex losses that are non-smooth nor strongly convex, the increase in the second moment compared to the usual stochastic gradients should be no greater than $\log_2(m)^{1/2}$). How to best refine our technique to better take into account the discrete structure of the output space is an open question. Introducing bias that does not modify convergence properties while reducing variance eventually thanks to importance sampling is a potential way to approach this problem. A simpler idea would be to remember information of the type $Y_i \in s$ to restrict the questions asked in order to locate $f_{\theta_t}(X_i) - Y_i$ when performing stochastic gradient descent with resampling. Combinatorial bandits might also provide helpful insights on the matter. Ultimately, we would like to build an understanding of the whole distribution $(Y \mid X)$ and not only of $f^*(X)$ as we explore labels in order to refine this exploration.

## 7.3 Min-max approaches

In this section, we discuss potential extensions of our SGD procedure, based on min-max variational objectives.

Min-max approaches have been popularized for searching games and active learning, where one searches for the question that minimizes the size of the space where a potential guess could lie under the worst possible answer to that question. A particularly well illustrative example is the solution of the Mastermind game proposed by Knuth (1977). While our work leverages plain SGD, one could build on the vector field point-of-view of gradient descent (see, *e.g.*, Bubeck, 2015) to tackle min-max convex concave problems with similar guarantees. In particular, we could design weakly supervised losses $L(f(x), s; \mathbf{1}_{y \in s})$ and min-max games where a prediction player aims at minimizing such a loss with respect to the prediction $f$, while the query player aims at maximizing it with respect to the question $s$, that is querying information that best elicit mistakes made by the prediction player. For example, the dual norm characterization of the norm leads to the following min-max approach to the median regression

$$\arg\min_{f:\mathcal{X}\to\mathcal{Y}} \mathcal{R}(f) = \arg\min_{f:\mathcal{X}\to\mathcal{Y}} \max_{U \in (\mathbb{S}^{m-1})^{\mathcal{X}\times\mathcal{Y}}} \mathbb{E}_{(X,Y)\sim\rho} \left[ \langle U(x, y), f(x) - y \rangle \right].$$

Such min-max formulations would be of interest if they lead to improvement of computational and statistical efficiencies, similarly to the work of Babichev et al. (2019). For classification problems, the following proposition introduces such a game and suggests its suitability. Its proof can be found in Appendix D.

**Proposition 4.** *Consider the classification problem of learning $f^* : \mathcal{X} \to \mathcal{Y}$ where $\mathcal{Y}$ is of finite cardinality, with the 0-1 loss $\ell(z, y) = \mathbf{1}_{z \neq y}$, minimizing the risk (1) under a distribution $\rho$ on $\mathcal{X} \times \mathcal{Y}$. Introduce the surrogate score functions $g : \mathcal{X} \to \Delta_{\mathcal{Y}}; x \to v$ where $v = (v_y)_{y \in \mathcal{Y}}$ is a family of non-negative weights that sum to one, as well as the surrogate loss function $L : \Delta_{\mathcal{Y}} \times \mathcal{S} \times \{-1, 1\} \to \mathbb{R}; (v, S, \varepsilon) = \varepsilon(1 - 2\sum_{y \in S} v_y)$, and the min-max game*

$$\min_{g:\mathcal{X}\to\Delta_{\mathcal{Y}}} \max_{\mu:\mathcal{X}\to\Delta_{\mathcal{S}}} \mathbb{E}_{(X,Y)\sim\rho} \mathbb{E}_{S\sim\mu(x)} \left[ L(g(x), S; \mathbf{1}_{Y \in S} - \mathbf{1}_{Y \notin S}) \right]. \tag{8}$$

*When $\mathcal{S}$ contains the singletons and with the low-noise condition that $\mathbb{P}(Y \neq f^*(x) \mid X = x) < 1/2$ almost everywhere, then $f^*$ can be learned through the relation $f^*(x) = \arg\min_{y \in \mathcal{Y}} g^*(x)_y$ for the unique minimizer $g^*$ of (8). Moreover, the minimization of the empirical version of this objective with the stochastic gradient updates for saddle point problems provides a natural "active labeling" scheme to find this $g^*$.*

On the one hand, this min-max formulation could help to easily incorporate restrictions on the sets to query. On the other hand, it is not completely clear how to best update (or derive an unbiased stochastic gradient strategy for) the adversarial query strategy $\mu$ based on partial information.

## 8 Conclusion

We have introduced the "active labeling" problem, which corresponds to "active partially supervised learning". We provided a solution to this problem based on stochastic gradient descent. Although our method can be used for any discrete output problem, we detailed how it works for median regression, where we show that it optimizes the generalization error for a given annotation budget. In a near future, we would like to focus on better exploiting the discrete structure of classification problems, eventually with resampling strategies.

Understanding more precisely the key issues in applications concerned with privacy, and studying how weak gradients might provide a good trade-off between learning efficiently and revealing too much information also provide interesting follow-ups. Finally, regarding dataset annotation, exploring different paradigms of weakly supervised learning would lead to different active weakly supervised learning frameworks. While this work is based on partial labeling, similar formalization could be made based on other weak supervision models, such as aggregation (*e.g.*, Ratner et al., 2020), or group statistics (Dietterich et al., 1997). In particular, annotating a huge dataset is often done by bagging inputs according to predicted labels and correcting errors that can be spotted on those bags of inputs (Deng et al., 2009). We left for future work the study of variants of the "active labeling" problem that model those settings.

## Acknowledgments and Disclosure of Funding

While at INRIA / ENS / PSL, VC was funded in part by the French government under management of Agence Nationale de la Recherche as part of the "Investissements d'avenir" program, reference ANR-19-P3IA-0001 (PRAIRIE 3IA Institute). FR and AR also acknowledges support of the European Research Council (grants SEQUOIA 724063 and REAL 947908).

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
