# A  Proofs of the statistical analysis

In the following proofs, we assume $\mathcal{X}$ to be Polish and $\mathcal{Y} = \mathbb{R}^m$, so to define the joint probability $\rho \in \Delta_{\mathcal{X} \times \mathcal{Y}}$. Moreover, we assume that $\mathbb{E}[\|Y\|] < +\infty$ in order to define the risk of median regression. We consider $\mathcal{H}$ to be a Hilbert space that is separable (*i.e.* only the origin is in all the neighborhood of the origin), and $\varphi$ to be a measurable mapping from $\mathcal{X}$ to $\mathcal{H}$.

In terms of notations, we denote $\{1, 2, \cdots, n\}$ by $[n]$ for any $n \in \mathbb{N}^*$, and by $(x_i)_{i \le n}$ the family $(x_1, \cdots, x_n)$ for any sequence $(x_i)$. The unit sphere in $\mathbb{R}^m$ is denoted by $\mathbb{S}^{m-1}$. The symbol $\otimes$ denotes tensors, and is extended to product measures in the notation $\rho^{\otimes n} = \rho \times \rho \times \cdots \times \rho$. We have used the isometry between trace-class linear mappings from $\mathcal{H}$ to $\mathcal{Y}$ and the tensor space $\mathcal{Y} \otimes \mathcal{H}$, which generalizes the matrix representation of linear map between two finite-dimensional vector spaces. This space inherits from the Hilbertian structure of $\mathcal{H}$ and $\mathcal{Y}$ and we denote by $\|\cdot\|$ the Hilbertian norm that generalizes the Frobenius norm on linear maps between Euclidean spaces.

## A.1  Upper bound for stochastic gradient descent

This subsection is devoted to the proof of Theorem 1. For simplicity, we will work with the rescaled step size $\gamma_t := c_2 \gamma(t)$ rather than the step size described in the main text $\gamma(t)$.

Convergence of stochastic gradient descent for non-smooth problems is a known result. For completeness, we reproduce and adapt a usual proof to our setting. For $t \in \mathbb{N}$, let us introduce the random functions

$$\mathcal{R}_t(\theta) = c_2^{-1} |\langle \theta \varphi(X_t) - Y_t, U_t \rangle|, \qquad \text{where} \qquad c_2 = \mathbb{E}_U[|\langle e_1, U \rangle|] = \mathbb{E}_U[\text{sign}(\langle e_1, U \rangle) \langle e_1, U \rangle]$$

for $(X_t, Y_t) \sim \rho$, $U_t$ uniform on the sphere $\mathbb{S}^{m-1} \subset \mathcal{Y}$. Those random functions all average to $\mathcal{R}(\theta) = \mathbb{E}_\rho \mathbb{E}_U[c_2^{-1} |\langle \theta \varphi(X) - Y, U \rangle|] = \mathbb{E}_\rho[\|\theta \varphi(X) - Y\|]$. After a random initialization $\theta_0 \in \Theta$, the stochastic gradient update rule can be written for any $t \in \mathbb{N}$ as

$$\theta_{t+1} = \theta_t - \gamma_t \nabla \mathcal{R}_t(\theta_t),$$

where $\nabla \mathcal{R}_t$ denotes any sub-gradients of $\mathcal{R}_t$. We can compute

$$\nabla \mathcal{R}_t(\theta_t) = c_2^{-1} \nabla |\langle \theta \varphi(X_t) - Y_t, U_t \rangle| = c_2^{-1} \text{sign}(\langle \theta \varphi(X_t) - Y_t, U_t \rangle) U_t \otimes \varphi(X_t).$$

This corresponds to the gradient written in Algorithm 1.

Let us now express the recurrence relation on $\|\theta_{t+1} - \theta^*\|$. We have

$$\begin{aligned}
\|\theta_{t+1} - \theta^*\|^2 &= \|\theta_t - \gamma_t \nabla \mathcal{R}_t(\theta_t) - \theta^*\|^2 \\
&= \|\theta_t - \theta^*\|^2 + \gamma_t^2 \|\nabla \mathcal{R}_t(\theta_t)\|^2 - 2\gamma_t \langle \nabla \mathcal{R}_t(\theta_t), \theta_t - \theta^* \rangle.
\end{aligned}$$

Because $\mathcal{R}_t$ is convex, it is above its tangents

$$\mathcal{R}_t(\theta^*) \ge \mathcal{R}_t(\theta_t) + \langle \nabla \mathcal{R}_t(\theta_t), \theta^* - \theta_t \rangle.$$

Hence,

$$\|\theta_{t+1} - \theta^*\|^2 \le \|\theta_t - \theta^*\|^2 + \gamma_t^2 \|\nabla \mathcal{R}_t(\theta_t)\|^2 + 2\gamma_t(\mathcal{R}_t(\theta^*) - \mathcal{R}_t(\theta_t)).$$

This allows bounding the excess of risk as

$$2(\mathcal{R}_t(\theta_t) - \mathcal{R}_t(\theta^*)) \le \frac{1}{\gamma_t}(\|\theta_t - \theta^*\|^2 - \|\theta_{t+1} - \theta^*\|^2) + \gamma_t c_2^{-2} \|\varphi(X_t)\|^2.$$

where we used the fact that $\|\nabla \mathcal{R}_t\| = c_2^{-1} \|\varphi(X_t)\|$. Let us multiply this inequality by $\eta_t > 0$ and sum from $t = 0$ to $t = T - 1$, we get

$$\begin{aligned}
2\left(\sum_{t=0}^{T-1} \eta_t \mathcal{R}_t(\theta_t) - \sum_{t=0}^{T-1} \eta_t \mathcal{R}_t(\theta^*)\right) &\le \sum_{t=0}^{T-1} \frac{\eta_t}{\gamma_t}(\|\theta_t - \theta^*\|^2 - \|\theta_{t+1} - \theta^*\|^2) + \sum_{t=0}^{T-1} \eta_t \gamma_t c_2^{-2} \|\varphi(X_t)\|^2 \\
&= \frac{\eta_0}{\gamma_0} \|\theta_0 - \theta^*\|^2 - \frac{\eta_{T-1}}{\gamma_{T-1}} \|\theta_T - \theta^*\|^2 + \sum_{t=1}^{T-1} \left(\frac{\eta_t}{\gamma_t} - \frac{\eta_{t-1}}{\gamma_{t-1}}\right) \|\theta_t - \theta^*\|^2 + \sum_{t=0}^{T-1} \eta_t \gamma_t c_2^{-2} \|\varphi(X_t)\|^2.
\end{aligned}$$

From here, there is several options to obtain a convergence result, either one assume $\|\theta_t - \theta^*\|$ bounded and take $\eta_t \gamma_{t-1} \geq \eta_{t-1} \gamma_t$; or one take $\eta_t = \gamma_t$ but at the price of paying an extra $\log(T)$ factor in the bound; or one take $\gamma_t$ and $\eta_t$ independent of $t$. Since we suppose the annotation budget given, we will choose $\gamma_t$ and $\eta_t$ independent of $t$, only depending on $T$.

$$2\left(\sum_{t=0}^{T-1} \eta \mathcal{R}_t(\theta_t) - \sum_{t=0}^{T-1} \eta \mathcal{R}_t(\theta^*)\right) \leq \frac{\eta}{\gamma} \|\theta_0 - \theta^*\|^2 + \sum_{t=0}^{T-1} \eta\gamma c_2^{-2} \|\varphi(X_t)\|^2 .$$

Let now take the expectation with respect to all the random variables, for the risk

$$\mathbb{E}_{(X_s,Y_s,U_s)_{s\leq t}}[\mathcal{R}_t(\theta_t)] = \mathbb{E}_{(X_s,Y_s,U_s)_{s\leq t}}\left[\mathbb{E}_{(X_t,Y_t)}\left[\mathbb{E}_{U_t}\left[\mathcal{R}_t(\theta_t)\,|\,\theta_t\right]\,|\,\theta_t\right]\right]$$
$$= \mathbb{E}_{(X_s,Y_s,U_s)_{s\leq t}}[\mathcal{R}(\theta_t)] = \mathbb{E}[\mathcal{R}(\theta_t)].$$

For the variance, $\mathbb{E}[\|\varphi(X_s)\|^2] = \mathbb{E}[\|\varphi(X)\|^2] = \kappa^2$.

Let us fix $T$ and consider $\eta_t = 1/T$, by Jensen we can bound the following averaging

$$2\left(\mathcal{R}\left(\sum_{t=0}^{T-1} \eta_t \theta_t\right) - \mathcal{R}(\theta^*)\right) \leq 2\left(\sum_{t=0}^{T-1} \eta_t \mathcal{R}(\theta_t) - \mathcal{R}(\theta^*)\right) = 2\,\mathbb{E}\left[\sum_{t=0}^{T-1} \eta_t(\mathcal{R}_t(\theta_t) - \mathcal{R}_t(\theta^*))\right]$$
$$\leq \frac{1}{T\gamma} \|\theta_0 - \theta^*\|^2 + \gamma c_2^{-2}\kappa^2.$$

Initializing $\theta_0$ to zero, we can optimize the resulting quantity to get the desired result.

### A.2 Upper bound for resampling strategy

For resampling strategies, the proof is built on classical statistical learning theory considerations. Let us decompose the risk between estimation and optimization errors. Recall the expression of the risk $\mathcal{R}$, the function taking as inputs measurable functions from $\mathcal{X}$ to $\mathcal{Y}$ and outputting a real number

$$\mathcal{R}(f) = \mathbb{E}_\rho[\|f(X) - Y\|].$$

Let us denote by $\mathcal{F}$ the class of functions from $\mathcal{X}$ to $\mathcal{Y}$ we are going to work with. Let $f_n$ be our estimate of $f^*$ which maps almost every $x \in \mathcal{X}$ to the geometric median of $(Y\,|\,X)$. Denote by $\mathcal{R}^*_{\mathcal{D}_n}$ the best value that can be achieved by our class of functions to minimize the empirical average absolute deviation

$$\mathcal{R}^*_{\mathcal{D}_n} = \inf_{f \in \mathcal{F}} \mathcal{R}_{\mathcal{D}_n}(f).$$

Assumption 1 states that we have a well-specified model $\mathcal{F}$ to estimate the median, *i.e.* $f^* \in \mathcal{F}$. Hence, the excess of risk can be decomposed as an estimation and an optimization error, without approximation error (it is not difficult to add an approximation error, but it will make the derivations longer and the convergence rates harder to parse for the reader). Using the fact that $\mathcal{R}_{\mathcal{D}_n}(f^*) \geq \mathcal{R}^*_{\mathcal{D}_n}$ by definition of the infimum, we have

$$\mathcal{R}(f_n) - \mathcal{R}(f^*) \leq \underbrace{\mathcal{R}(f_n) - \mathcal{R}_{\mathcal{D}_n}(f_n) + \mathcal{R}_{\mathcal{D}_n}(f^*) - \mathcal{R}(f^*)}_{\text{estimation error}} + \underbrace{\mathcal{R}_{\mathcal{D}_n}(f_n) - \mathcal{R}^*_{\mathcal{D}_n}}_{\text{optimization error}}. \tag{9}$$

**Estimation error.**   Let us begin by controlling the estimation error. We have two terms in it. $\mathcal{R}_{\mathcal{D}_n}(f^*) - \mathcal{R}(f^*)$ can be controlled with a concentration inequality on the empirical average of $\|f^*(X) - Y\|$ around its population mean. Assuming sub-Gaussian moments of $Y$, it can be done with Bernstein inequality.

$\mathcal{R}_{\mathcal{D}_n}(f_n) - \mathcal{R}(f_n)$ is harder to control as $f_n$ depends on $\mathcal{D}_n$, so we can not use the same technique. The classical technique consists in going for the brutal uniform majoration,

$$\mathcal{R}(f_n) - \mathcal{R}_{\mathcal{D}_n}(f_n) \leq \sup_{f \in \mathcal{F}} \left(\mathcal{R}(f) - \mathcal{R}_{\mathcal{D}_n}(f)\right), \tag{10}$$

where $\mathcal{F}$ denotes the set of functions that $f_n$ could be in concordance with our algorithm. While this bound could seem highly suboptimal, when the class of functions is well-behaved, we can indeed control the deviation $\mathcal{R}(f) - \mathcal{R}_{\mathcal{D}_n}(f)$ uniformly over this class without losing much (indeed for any

class of functions, it is possible to build some really adversarial distribution $\rho$ so that this supremum behaves similarly to the concentration we are looking for (Vapnik, 1995; Anthony and Bartlett, 1999)). This is particularly the case for our model linked with Assumption 1. Expectations of supremum processes have been extensively studied, allowing to get satisfying upper bounds (note that when the $\|f(X) - Y\|$ is bounded, deviation of the quantity of interest around its expectation can be controlled through McDiarmid inequality). In the statistical learning literature, it is usual to proceed with Rademacher complexity.

**Lemma 5** (Uniform control of functions deviation with Rademacher complexity). *The expectation of the excess of risk can be bounded as*

$$\frac{1}{2} \mathbb{E}_{\mathcal{D}_n} \left[ \sup_{f \in \mathcal{F}} \left( \mathcal{R}(f) - \mathcal{R}_{\mathcal{D}_n}(f) \right) \right] \leq \Re_n(\mathcal{F}, \ell, \rho) := \frac{1}{n} \mathbb{E}_{\mathcal{D}_n, (\sigma_i)} \left[ \sup_{f \in \mathcal{F}} \sigma_i \ell(f(X_i), Y_i) \right], \qquad (11)$$

*where $(\sigma_i)_{i \leq n}$ is defined as a family of Bernoulli independent variables taking value one or minus one with equal probability, and $\Re_n(\mathcal{F}, \ell, \rho)$ is called Rademacher complexity.*

*Proof.* This results from the reduction to larger supremum and a symmetrization trick,

$$\mathbb{E}_{\mathcal{D}_n} \left[ \sup_{f \in \mathcal{F}} \left( \mathcal{R}(f) - \mathcal{R}_{\mathcal{D}_n}(f) \right) \right] = \mathbb{E}_{\mathcal{D}_n} \left[ \sup_{f \in \mathcal{F}} \left( \mathbb{E}_{\mathcal{D}'_n} \mathcal{R}_{\mathcal{D}'_n}(f) - \mathcal{R}_{\mathcal{D}_n}(f) \right) \right]$$

$$\leq \mathbb{E}_{\mathcal{D}_n} \mathbb{E}_{\mathcal{D}'_n} \left[ \sup_{f \in \mathcal{F}} \left( \mathcal{R}_{\mathcal{D}'_n}(f) - \mathcal{R}_{\mathcal{D}_n}(f) \right) \right]$$

$$= \mathbb{E}_{(X_i, Y_i), (X'_i, Y'_i)} \left[ \sup_{f \in \mathcal{F}} \left( \frac{1}{n} \sum_{i=1}^{n} \ell(f(X'_i), Y'_i) - \ell(f(X_i), Y_i) \right) \right]$$

$$= \mathbb{E}_{(X_i, Y_i), (X'_i, Y'_i), (\sigma_i)} \left[ \sup_{f \in \mathcal{F}} \left( \frac{1}{n} \sum_{i=1}^{n} \sigma_i \left( \ell(f(X'_i), Y'_i) - \ell(f(X_i), Y_i) \right) \right) \right]$$

$$\leq 2 \mathbb{E}_{(X_i, Y_i), (\sigma_i)} \left[ \sup_{f \in \mathcal{F}} \left( \frac{1}{n} \sum_{i=1}^{n} \sigma_i \left( \ell(f(X_i), Y_i) \right) \right) \right],$$

which ends the proof. $\qquad \square$

In our case, we want to compute the Rademacher complexity for $\ell$ given by the norm of $\mathcal{Y}$, and $\mathcal{F} = \{x \to \theta \varphi(x) \mid \theta \in \mathcal{Y} \otimes \mathcal{H}, \|\theta\| < M\}$, for $M > 0$ a parameter to specify in order to make sure that $\|\theta^*\| < M$, where the norm has to be understood as the $\ell^2$-product norm on $\mathcal{Y} \otimes \mathcal{H} \simeq \mathcal{H}^m$. Working with linear models and Lipschitz losses is a well-known setting, allowing to derive directly the following bound.

**Lemma 6** (Rademacher complexity of linear models with Lipschitz losses). *The complexity of the linear class of vector-valued function $\mathcal{F} = \{x \to \theta \varphi(x) \mid \theta \in \mathcal{Y} \otimes \mathcal{H}, \|\theta\| < M\}$ is bounded as*

$$\mathbb{E}_{(\sigma_i)} \left[ \sup_{f \in \mathcal{F}} \left( \frac{1}{n} \sum_{i=1}^{n} \sigma_i \|f(x_i) - y_i\| \right) \right] \leq M m^{1/2} \kappa n^{-1/2}. \qquad (12)$$

*Proof.* This proposition is usually split in two. First using the fact that the composition of a space of functions with a Lipschitz function does not increase the entropy of the subsequent space (Vitushkin, 1954). Then bounding the Rademacher complexity of linear models. We refer to Maurer (2016) for a self-contained proof of this result (stated in its Section 4.3). $\qquad \square$

Adding all the pieces together we have proven the following proposition, using the fact that the previous bound also applies to $\sup_{f \in \mathcal{F}} \mathcal{R}_{\mathcal{D}_n}(f) - \mathcal{R}(f)$ by symmetry, hence it can be used for the deviation of $\mathcal{R}_{\mathcal{D}_n}(f^*) - \mathcal{R}(f^*)$.

**Proposition 7** (Control of the estimation error). *Under Assumption 1, with the model of computation $\mathcal{F} = \{x \in \mathcal{X} \to \theta\varphi(x) \in \mathcal{Y} \mid \|\theta\| \leq M\}$, the generalization error of $f_n$ is controlled by a term in $n^{-1/2}$ plus an optimization error on the empirical risk minimization*

$$\mathbb{E}_{\mathcal{D}_n}\left[\mathcal{R}(f_n) - \mathcal{R}(f^*)\right] \leq \frac{4m^{1/2}M\kappa}{n^{1/2}} + \mathbb{E}_{\mathcal{D}_n}\left[\mathcal{R}_{\mathcal{D}_n}(f_n) - \mathcal{R}^*_{\mathcal{D}_n}\right], \tag{13}$$

*as long as $f^* \in \mathcal{F}$.*

Note that this result can be refined using regularized risk (Sridharan et al., 2008), which would be useful under richer (stronger or weaker) source assumptions (*e.g.*, Caponnetto and De Vito, 2006). Such a refinement would allow switching from a constraint $\|\theta\| < M$ to define $\mathcal{F}$ to a regularization parameter $\lambda\|\theta\|^2$ added in the risk without restrictions on $\|\theta\|$, which would be better aligned with the current practice of machine learning. Under Assumption 1, this will not fundamentally change the result. The estimation error can be controlled with the derivation in Appendix A.1, where stochastic gradients correspond to random sampling of a coefficient $i_t \leq n$ plus the choice of a random $U_t$. For the option without resampling, there exists an acceleration scheme specific to different losses in order to benefit from the strong convexity (*e.g.*, Bach and Moulines, 2013).

## A.3 Lower bound

In this section, we prove Theorem 2. Let us consider any algorithm $\mathcal{A} : \cup_{n\in\mathbb{N}}(\mathcal{X} \times \mathcal{Y})^n \to \Theta$ that matches a dataset $\mathcal{D}_n$ to an estimate $\theta_{\mathcal{D}_n} \in \Theta$. Let us consider jointly a distribution $\rho$ and a parameter $\theta$ such that Assumption 1 holds, that is $f_\rho := \arg\min_{f:\mathcal{X}\to\mathcal{Y}} \mathbb{E}_\rho[\ell(f(X), Y)] = f_\theta$. We are interested in characterizing for each algorithm the worst excess of risk it can achieve with respect to an adversarial distribution. The best worst performance that can be achieved by algorithms matching datasets to parameter can be written as

$$\mathcal{E} = \inf_{\mathcal{A}} \sup_{\theta\in\Theta, \rho\in\Delta_{\mathcal{X}\times\mathcal{Y}}; f_\rho=f_\theta} \mathbb{E}_{\mathcal{D}_n\sim\rho^{\otimes n}}\left[\mathbb{E}_{(X,Y)\sim\rho}\left[\ell(f_{\mathcal{A}(\mathcal{D}_n)}(X), Y) - \ell(f_\theta(X), Y)\right]\right]. \tag{14}$$

This provides a lower bound to upper bounds such as (6) that can be derived for any algorithm. There are many ways to get lower bounds on this quantity. Ultimately, we want to quantify the best certainty one can have on an estimate $\theta$ based on some observations $(X_i, Y_i)_{i\leq n}$. In particular, the algorithms $\mathcal{A}$ can be seen as rules to discriminate a model $\theta$ from observations $\mathcal{D}_n$ made under $\rho_\theta$, and where the error is measured through the excess of risk $\mathcal{R}(f_{\hat{\theta}}, \rho_\theta) - \mathcal{R}(f_\theta; \rho_\theta)$ where $\mathcal{R}(f; \rho) = \mathbb{E}_\rho[\ell(f(X), Y)]$ and $\rho_\theta$ is a distribution parametrized by $\theta$ such that $f_\theta = f_\rho$.

Let us first characterize the measure of error. Surprisingly, when in presence of Gaussian noise or uniform noise, the excess of risk behaves like a quadratic metric between parameters.

**Lemma 8** (Quadratic behavior of the median regression excess of risk with Gaussian noise). *Consider the random variable $Y \sim \mathcal{N}(\mu, \sigma^2 I_m)$, denote by $\hat{\mu}$ an estimate of $\mu$, the excess of risk can be developed as*

$$\mathbb{E}_{\mathcal{N}(\mu,\sigma^2 I_m)}[\|\hat{\mu} - Y\| - \|\mu - Y\|] = \frac{c_4\|\hat{\mu} - \mu\|^2}{\sigma} + o\left(\frac{\|\hat{\mu} - \mu\|^3}{\sigma^2}\right), \tag{15}$$

*where $c_4 = \Gamma(\frac{m+1}{2})/(2\sqrt{2}\Gamma(\frac{m+2}{2})) \geq (m+2)^{-1/2}/2$.*

*Proof.* With this specific noise model, one can do the following derivations.

$$\mathbb{E}_{\mathcal{N}(\mu,\sigma^2 I_m)}[\|\hat{\mu} - Y\|] = \mathbb{E}_{\mathcal{N}(0,I_m)}[\|\hat{\mu} - \mu - \sigma Y\|] = \sigma\,\mathbb{E}_{\mathcal{N}(0,I_m)}\left[\left\|\frac{\hat{\mu} - \mu}{\sigma} - Y\right\|\right].$$

We recognize the mean of a non-central $\chi$-distribution of parameter $k = m$ and $\lambda = \left\|\frac{\hat{\mu}-\mu}{\sigma}\right\|$. It can be expressed through the generalized Laguerre functions, which allows us to get the following Taylor expansion

$$\mathbb{E}_{\mathcal{N}(\mu,\sigma^2 I_m)}[\|\hat{\mu} - Y\|] = \frac{\sqrt{\pi}\sigma}{\sqrt{2}} L_{\frac{1}{2}}^{(\frac{m-2}{2})}\left(-\frac{\|\hat{\mu} - \mu\|^2}{2\sigma^2}\right)$$

$$= \frac{\sqrt{\pi}\sigma}{\sqrt{2}}\left(L_{\frac{1}{2}}^{(\frac{m-2}{2})}(0) + \frac{\|\hat{\mu} - \mu\|^2}{2\sigma^2} L_{-\frac{1}{2}}^{(\frac{m}{2})}(0)\right) + o\left(\frac{\|\hat{\mu} - \mu\|^3}{\sigma^2}\right).$$

Hence, the following expression of the excess of risk,

$$\mathbb{E}_{\mathcal{N}(\mu,\sigma^2 I_m)}[\|\hat{\mu} - Y\| - \|\mu - Y\|] = \frac{\sqrt{\pi}\,\|\hat{\mu} - \mu\|^2}{2\sqrt{2}\sigma} L_{-\frac{1}{2}}^{(\frac{m}{2})}(0) + o\left(\frac{\|\hat{\mu} - \mu\|^3}{\sigma^2}\right)$$

$$= \frac{\Gamma(\frac{m+1}{2})\,\|\hat{\mu} - \mu\|^2}{2\sqrt{2}\Gamma(\frac{m+2}{2})\sigma} + o\left(\frac{\|\hat{\mu} - \mu\|^3}{\sigma^2}\right).$$

Note that in dimension one, the calculation can be done explicitly by computing integrals with the error function.

$$\mathbb{E}_{\mathcal{N}(\mu,\sigma^2)}[\|\hat{\mu} - Y\|] = \sigma\,\mathbb{E}_{\mathcal{N}(0,1)}\left[Y - \frac{\hat{\mu} - \mu}{\sigma} + 2\mathbf{1}_{Y < \frac{\hat{\mu}-\mu}{\sigma}}\left(\frac{\hat{\mu} - \mu}{\sigma} - Y\right)\right]$$

$$= \mu - \hat{\mu} + 2(\hat{\mu} - \mu)\,\mathbb{E}_{\mathcal{N}(0,1)}\left[\mathbf{1}_{Y < \frac{\hat{\mu}-\mu}{\sigma}}\right] - 2\sigma\,\mathbb{E}_{\mathcal{N}(0,1)}\left[Y\mathbf{1}_{Y < \frac{\hat{\mu}-\mu}{\sigma}}\right]$$

$$= \mu - \hat{\mu} + 2(\hat{\mu} - \mu)\left(\frac{1}{2} + \frac{1}{2}\operatorname{erf}\left(\frac{\hat{\mu} - \mu}{\sqrt{2}\sigma}\right)\right) - \frac{\sqrt{2}\sigma}{\sqrt{\pi}}\int_{-\infty}^{\frac{\hat{\mu}-\mu}{\sigma}} y e^{-\frac{y^2}{2}}\,\mathrm{d}y$$

$$= (\hat{\mu} - \mu)\operatorname{erf}\left(\frac{\hat{\mu} - \mu}{\sqrt{2}\sigma}\right) - \frac{\sqrt{2}\sigma}{\sqrt{\pi}}e^{-\frac{(\hat{\mu}-\mu)^2}{2\sigma^2}},$$

where we used the error function, which is the symmetric function defined for $x \in \mathbb{R}_+$ as

$$\operatorname{erf}(x) = \frac{2}{\sqrt{\pi}}\int_0^x e^{-t^2}\,\mathrm{d}t = \frac{2}{\sqrt{2\pi}}\int_0^{\sqrt{2}x} e^{-\frac{u^2}{2}}\,\mathrm{d}u = 2\,\mathbb{E}_{\mathcal{N}(0,1)}[\mathbf{1}_{0 \leq Y \leq \sqrt{2}x}].$$

Developing those two functions in the Taylor series leads to the same quadratic behavior. □

Let us now add a context variable.

**Lemma 9** (Reduction to least-squares). *For $\mathcal{Y} = \mathbb{R}^m$, there exists a $\sigma_m > 0$, such that if $\varphi$ is bounded by $\kappa$, and $f^*$ belongs to the class of functions $\mathcal{F} = \{x \to \theta\varphi(x) \mid \theta \in \mathcal{Y} \otimes \mathcal{H}, \|\theta\| \leq M\}$, and the conditional distribution are distributed as $(Y \mid X) \sim \mathcal{N}(f^*(x), \sigma^2 I_m)$, with $\sigma > 2M\kappa\sigma_m$,*

$$\forall f \in \mathcal{F}, \qquad \mathcal{R}(f) - \mathcal{R}(f^*) \geq \frac{c_4\,\|f - f^*\|_{L^2(\rho_{\mathcal{X}})}^2}{2\sigma}. \tag{16}$$

*Proof.* According to the precedent lemma, there exists $\sigma_m$ such that $\|\hat{\mu} - \mu\|\,\sigma^{-1} \leq \sigma_m^{-1}$ leads to[5]

$$\mathbb{E}_{\mathcal{N}(\mu,\sigma^2 I_m)}[\|\hat{\mu} - Y\| - \|\mu - Y\|] \geq \frac{c_4\,\|\hat{\mu} - \mu\|^2}{2\sigma}.$$

Let $f$ and $f^* \in \mathcal{F}$ be parametrized by $\theta$ and $\theta^*$. For a given $x$, setting $\hat{\mu} = f_\theta(x) = \theta\varphi(x)$ and $\mu = f_{\theta^*}(x)$, we get that, using the operator norm,

$$\|\hat{\mu} - \mu\| = \|(\theta - \theta^*)\varphi(x)\| \leq \|\theta - \theta^*\|_{\mathrm{op}}\|\varphi(x)\| \leq \|\theta - \theta^*\|\,\|\varphi(x)\| \leq 2M\kappa.$$

Hence, as soon as $2M\kappa \leq \sigma\sigma_m^{-1}$, we have that for almost all $x \in \mathcal{X}$

$$\mathbb{E}_Y\left[\|f(X) - Y\| - \|f^*(X) - Y\| \mid X = x\right] \geq \frac{c_4\,\|f(X) - f^*(X)\|^2}{2\sigma}.$$

The result follows from integration over $\mathcal{X}$. □

We now have a characterization of the excess of risk that will allow us to reuse lower bounds for least-squares regression. We will follow the exposition of Bach (2023) that we reproduce and comment here for completeness. It is based on the generalized Fano's method (Ibragimov and Khas'minskii, 1977; Birgé, 1983).

Learnability over a class of functions depends on the size of this class of functions. For least-squares regression with a Hilbert class of functions, the right notion of size is given by the Kolmogorov

---

[5]This best value for $\sigma_m$ can be derived by studying the Laguerre function, which we will not do in this paper.

entropy. Let us call $\varepsilon$-packing of $\mathcal{F}$ with a metric $d$ any family $(f_i)_{i\leq N} \in \mathcal{F}^N$ such that $d(f_i, f_j) > \varepsilon$. The logarithm of the maximum cardinality of an $\varepsilon$-packing defines the $\varepsilon$-capacity of the class of functions $\mathcal{F}$. We refer the interested reader to Theorem 6 in Kolmogorov and Tikhomirov (1959) to make a link between the notions of capacity and entropy of a space. To be perfectly rigorous, the least-squares error in not a norm on the space of $L^2$ functions, but we will call it a *quasi-distance* as it verifies symmetry, positive definiteness and the inequality $d(x, y) \leq K(d(x, z) + d(z, y))$ for $K \geq 1$. Let us define an $\varepsilon$-packing with respect to a quasi-distance similarly as before.

The $\varepsilon$-capacity of a space $\mathcal{F}$ gives a lower bound on the number of information to transmit in order to recover a function in $\mathcal{F}$ up to precision $\varepsilon$. We will leverage this fact in order to show our lower bound. Let us first reduce the problem to a statistical test.

**Lemma 10** (Reduction to statistical testing). *Let us consider a class of functions $\mathcal{F}$ and an $\varepsilon$-packing $(f_i)_{i\leq N}$ of $\mathcal{F}$ with respect to a quasi-distance $d(\cdot, \cdot)$ verifying the triangular inequality up to a multiplicative factor $K$. Then the minimax optimality of an algorithm $\mathcal{A}$ that takes as input the dataset $\mathcal{D}_n = (X_i, Y_i)_{i\leq n}$ and output a function in $\mathcal{F}$ can be related to the minimax optimality of an algorithm $\mathcal{C}$ that takes an input the dataset $\mathcal{D}_n$ and output an index $j \in [m]$ through*

$$\inf_{\mathcal{A}} \sup_{\rho} \mathbb{E}_{\mathcal{D}_n \sim \rho^{\otimes n}} \left[ d\left( f_{\mathcal{A}(\mathcal{D}_n)}, f_\rho \right) \right] \geq \frac{\varepsilon}{2K} \inf_{\mathcal{C}} \sup_{i \in [N]} \mathbb{P}_{\mathcal{D}_n \sim (\rho_i)^{\otimes n}} \left( \mathcal{C}(\mathcal{D}_n) \neq i \right), \qquad (17)$$

*where the supremum over $\rho$ has to be understood as taken over all measures whose marginals can be written $\mathcal{N}(f^*(x), \sigma)$ for $\sigma$ bigger than a threshold $\sigma_m$ and $f^* \in \mathcal{F}$, and the supremum over $\rho_i$ taken over the same type of measures with $f^* \in (f_i)_{i\leq N}$.*

*Proof.* Consider an algorithm $\mathcal{A}$ that takes as input a dataset $\mathcal{D}_n = (X_j, Y_j)_{j\leq n}$ and output a function $f \in \mathcal{F}$. We would like to see $\mathcal{A}$ as deriving from a classification rule and relate the classification and regression errors. The natural classification rule associated with the algorithm $\mathcal{A}$ can be defined through $\pi$ the projection from $\mathcal{F}$ to $[N]$ that minimizes $d(f, f_{\pi(f)})$. The classification error and regression error made by $\pi \circ \mathcal{A}$ can be related thanks to the $\varepsilon$-packing property. For any index $j \in [N]$

$$d\left( f_{\pi \circ \mathcal{A}(\mathcal{D}_n)}, f_j \right) \geq \varepsilon \mathbf{1}_{\pi \circ \mathcal{A}(\mathcal{D}_n) \neq j}.$$

The error made by $f_{\mathcal{A}(\mathcal{D}_n)}$ relates to the one made by $f_{\pi \circ \mathcal{A}(\mathcal{D}_n)}$ thanks to the modified triangular inequality, using the definition of the projection

$$d\left( f_{\pi \circ \mathcal{A}(\mathcal{D}_n)}, f_j \right) \leq K \left( d\left( f_{\pi \circ \mathcal{A}(\mathcal{D}_n)}, f_{\mathcal{A}(\mathcal{D}_n)} \right) + d\left( f_{\mathcal{A}(\mathcal{D}_n)}, f_j \right) \right) \leq 2K d\left( f_{\mathcal{A}(\mathcal{D}_n)}, f_j \right).$$

Finally,

$$d\left( f_{\mathcal{A}(\mathcal{D}_n)}, f_j \right) \geq \frac{\varepsilon}{2K} \mathbf{1}_{\pi \circ \mathcal{A}(\mathcal{D}_n) \neq j}.$$

Assuming that the data were generated by a $\rho_i$ and taking the expectation, the supremum over $\rho_i$ and the infimum over $\mathcal{A}$ leads to

$$\inf_{\mathcal{A}} \sup_{\rho_i} \mathbb{E}_{\mathcal{D}_n \sim \rho_i^{\otimes n}} \left[ d\left( f_{\mathcal{A}(\mathcal{D}_n)}, f_i \right) \right] \geq \frac{\varepsilon}{2K} \inf_{\mathcal{C} = \pi \circ \mathcal{A}} \sup_{(\rho_i)} \mathbb{P}_{\mathcal{D}_n \sim \rho_i^{\otimes n}} \left( \mathcal{C}(\mathcal{D}_n) \neq i \right).$$

Because $\pi \circ \mathcal{A}$ are part of classification rules (indeed it parametrizes all the classification rules, simply consider $\mathcal{A}$ that matches a dataset to one of the functions $(f_i)_{i\leq N}$), and because the distributions $\rho_i$ are part of the distributions $\rho$ defined in the lemma, this last equation implies the stated result. $\quad\square$

One of the harshest inequalities in the last proof is due to the usage of the $\varepsilon$-packing condition without considering error made by $d\left( f_{\pi \circ \mathcal{A}(\mathcal{D}_n)}, f_j \right)$ that might be much worse than $\varepsilon$. We will later add a condition on the $\varepsilon$-packings to ensure that the $(f_i)$ are not too far from each other. This will not be a major problem when considering small balls in big dimension spaces.

### A.3.1 Results from statistical testing

In this section, we expand on lower bounds for statistical testing. We refer the curious reader to Cover and Thomas (1991). We begin by relaxing the supremum by an average

$$\inf_{\mathcal{C}} \sup_{i \in [N]} \mathbb{P}_{\mathcal{D}_n \sim (\rho_i)^{\otimes n}} \left( \mathcal{C}(\mathcal{D}_n) \neq i \right) = \inf_{\mathcal{C}} \sup_{p \in \Delta_N} \sum_{i=1}^{N} p_i \, \mathbb{P}_{\mathcal{D}_n \sim (\rho_i)^{\otimes n}} \left( \mathcal{C}(\mathcal{D}_n) \neq i \right) \qquad (18)$$

$$\geq \inf_{\mathcal{C}} \frac{1}{N} \sum_{i=1}^{N} \mathbb{P}_{\mathcal{D}_n \sim (\rho_i)^{\otimes n}} \left( \mathcal{C}(\mathcal{D}_n) \neq i \right). \tag{19}$$

The last quantity can be seen as the best measure of error that can be achieved by a decoder $\mathcal{C}$ of a signal $i \in [N]$ based on noisy observations $\mathcal{D}_n$ of the signal. A lower bound on such a similar quantity is the object of Fano's inequality (Fano, 1968).

**Lemma 11** (Fano's inequality). *Let $(X, Y)$ be a couple of random variables in $\mathcal{X} \times \mathcal{Y}$ with $\mathcal{X}$, $\mathcal{Y}$ finite, and $\hat{X} : \mathcal{Y} \to \mathcal{X}$ be a classification rule. Then, the error $e = e(X, Y) = \mathbf{1}_{X \neq \hat{X}(Y)}$ verifies*

$$H(X \,|\, Y) \leq H(e) + \mathbb{P}(e) \log(|\mathcal{X}| - 1) \leq \log(2) + \mathbb{P}(e) \log(|\mathcal{X}|).$$

*Where for $(X, Y) \in \Delta_{\mathcal{X} \times \mathcal{Y}}$, $H(X)$ and $H(X \,|\, Y)$ denotes the entropy and conditional entropy, defined as, with the convention $0 \log 0 = 0$,*

$$H(X) = - \sum_{x \in \mathcal{X}} \mathbb{P}(X = x) \log(\mathbb{P}(X = x)),$$

$$H(X \,|\, Y) = - \sum_{x \in \mathcal{X}, y \in \mathcal{Y}} \mathbb{P}(X = x, Y = y) \log(\mathbb{P}(X = x \,|\, Y = y)).$$

*Proof.* This lemma is actually the result of two properties. The first part of the proof is due to some manipulation of the entropy, consisting in showing that

$$H\left(X \,\big|\, \hat{X}(Y)\right) \leq H(e) + \mathbb{P}(e) \log(|\mathcal{X}| - 1). \tag{20}$$

Let us first recall the following additive property of entropy

$$H(X, Y \,|\, Z) = - \sum_{x \in \mathcal{X}, y \in \mathcal{Y}, z \in \mathcal{Z}} \mathbb{P}(X = x, Y = y, Z = z) \log(\mathbb{P}(X = x, Y = y \,|\, Z = z))$$

$$= - \sum_{x \in \mathcal{X}, y \in \mathcal{Y}, z \in \mathcal{Z}} \mathbb{P}(X = x, Y = y, Z = z) \log(\mathbb{P}(Y = y \,|\, X = x, Z = z))$$

$$- \sum_{x \in \mathcal{X}, y \in \mathcal{Y}, z \in \mathcal{Z}} \mathbb{P}(X = x, Y = y, Z = z) \log(\mathbb{P}(X = x \,|\, Z = z))$$

$$= H(Y \,|\, X, Z) + H(X \,|\, Z).$$

Using this chain rule, we get

$$H\left(e, X \,\big|\, \hat{X}\right) = H\left(e \,\big|\, X, \hat{X}\right) + H\left(X \,\big|\, \hat{X}\right)$$
$$= H\left(X \,\big|\, e, \hat{X}\right) + H\left(e \,\big|\, \hat{X}\right)$$

Because $e$ is a function of $\hat{X}$ and $X$ one can check that $H\left(e \,\big|\, X, \hat{X}\right) = 0$,

$$H\left(e \,\big|\, X, \hat{X}\right) = - \sum_{e, X, \hat{X}} \mathbb{P}(X, \hat{X}) \, \mathbb{P}\left(e \,\big|\, X, \hat{X}\right) \log(\mathbb{P}\left(e \,\big|\, X, \hat{X}\right))$$

$$= - \sum_{e, X, \hat{X}} \mathbb{P}(X, \hat{X}) \mathbf{1}_{e = \mathbf{1}_{X \neq \hat{X}}} \log(\mathbf{1}_{e = \mathbf{1}_{X \neq \hat{X}}}) = - \sum_{e, X, \hat{X}} \mathbb{P}(X, \hat{X}) \cdot 0 = 0.$$

Using Jensen inequality for the logarithm, we get

$$H\left(X \,\big|\, e, \hat{X}\right) = - \sum_{X, e, \hat{X}} \mathbb{P}(X, e, \hat{X}) \log(\mathbb{P}\left(X \,\big|\, e, \hat{X}\right))$$

$$= - \sum_{x, x'} \mathbb{P}(X = x, e = 0, \hat{X} = x') \log(\mathbb{P}\left(X = x \,\big|\, e = 0, \hat{X} = x'\right))$$

$$- \mathbb{P}(X = x, e = 1, \hat{X} = x') \log(\mathbb{P}\left(X = x \,\big|\, e = 1, \hat{X} = x'\right))$$

$$= - \sum_{x, x'} \mathbb{P}\left(X = x, \hat{X} = x'\right) \mathbf{1}_{x = x'} \log(\mathbf{1}_{x = x'})$$

$$- \mathbb{P}(e = 1) \mathbf{1}_{x \neq x'} \mathbb{P}(X = x, \hat{X} = x') \log(\mathbb{P}\left(X = x \,\big|\, \hat{X} = x'\right))$$

$$= \mathbb{P}(e = 1) \sum_{x'} \mathbb{P}(\hat{X} = x') \sum_{x \neq x'} \mathbb{P}\left(X = x \,\middle|\, \hat{X} = x'\right) \log\left(\frac{1}{\mathbb{P}\left(X = x \,\middle|\, \hat{X} = x'\right)}\right)$$

$$\leq \mathbb{P}(e = 1) \sum_{x'} \mathbb{P}(\hat{X} = x') \log\left(\sum_{x \neq x'} \mathbb{P}\left(X = x \,\middle|\, \hat{X} = x'\right) \frac{1}{\mathbb{P}\left(X = x \,\middle|\, \hat{X} = x'\right)}\right)$$

$$= \mathbb{P}(e = 1) \log(|\mathcal{X}| - 1).$$

Using that conditioning reduces the entropy, which follows again from Jensen inequality,

$$H(X) - H(X \,|\, Y) = \sum_{x,y} \mathbb{P}(X = x, Y = y) \log\left(\frac{\mathbb{P}(X = x \,|\, Y = y)}{\mathbb{P}(X = x)}\right)$$

$$= -\sum_{x,y} \mathbb{P}(X = x, Y = y) \log\left(\frac{\mathbb{P}(X = x)\,\mathbb{P}(Y = y)}{\mathbb{P}(X = x, Y = y)}\right)$$

$$\geq -\log\left(\sum_{x,y} \mathbb{P}(X = x, Y = y) \frac{\mathbb{P}(X = x)\,\mathbb{P}(Y = y)}{\mathbb{P}(X = x, Y = y)}\right) = 0,$$

we get

$$H\left(e \,\middle|\, \hat{X}\right) \leq H(e) \leq \log(2).$$

Hence, we have proven that

$$H\left(X \,\middle|\, \hat{X}\right) \leq \mathbb{P}(e = 1) \log(|\mathcal{X}| - 1) + H(e).$$

The rest of the proof follows from the so-called data processing inequality, that is

$$H\left(X \,\middle|\, \hat{X}(Y)\right) \geq H\left(X \,|\, Y\right). \tag{21}$$

We will not derive it here, since it will not be used in the following. $\qquad\square$

In our case, a slight modification of the proof of Fano's inequality leads to the following Proposition.

**Lemma 12** (Generalized Fano's method). *For any family of distributions $(\rho_i)_{i \leq N}$ on $\mathcal{X} \times \mathcal{Y}$ with $N \in \mathbb{N}^*$, any classification rule $\mathcal{C} : \mathcal{D}_n \to [N]$ cannot beat the following average lower bound*

$$\inf_{\mathcal{C}} \frac{1}{N} \sum_{i=1}^{N} \mathbb{P}_{\mathcal{D}_n \sim \rho_i^{\otimes n}}(\mathcal{C}(\mathcal{D}_n) \neq i) \log(N - 1) \geq \log(N) - \log(2) - \frac{n}{N^2} \sum_{i,j \in [N]} K\left(\rho_i \,\middle|\middle|\, \rho_j\right), \tag{22}$$

*where $K(p \,||\, q)$ is the Kullback-Leibler divergence defined for any measure $p$ absolutely continuous with respect to a measure $q$ as*

$$K(p \,||\, q) = \mathbb{E}_{X \sim q}\left[-\log\left(\frac{\mathrm{d}p(X)}{\mathrm{d}q(X)}\right)\right].$$

*Proof.* Let us consider the joint variable $(X, Y)$ where $X$ is a uniform variable on $[N]$ and $(Y \,|\, X)$ is distributed according to $\rho_X^{\otimes n}$. For any classification rule $\hat{X} : \mathcal{D}_n \to [N]$, using (20) we get

$$\frac{1}{N} \sum_{i=1}^{N} \mathbb{P}_{\mathcal{D}_n \sim \rho_i^{\otimes n}}\left(\hat{X}(\mathcal{D}_n) \neq i\right) = \mathbb{P}(\hat{X} \neq X) \log(N - 1) \geq H\left(X \,\middle|\, \hat{X}\right) - \log(2).$$

We should work on $H\left(X \,\middle|\, \hat{X} \,\middle|\, X\right)$ with similar ideas to the data processing inequality. First of all, using the chain rule for entropy

$$H\left(X \,\middle|\, \hat{X}\right) = H(X, \hat{X}) - H(\hat{X}) = H(X) + (H(X, \hat{X}) - H(X) - H(\hat{X})) = \log(N) - I(X, \hat{X}),$$

where $I$ is the mutual information defined as, for $X$ and $Z$ discrete

$$I(X, Z) = H(X) + H(Z) - H(X, Z) = \sum_{x,z} \mathbb{P}(X = x, Z = z) \log\left(\frac{\mathbb{P}(X = x, Z = z)}{\mathbb{P}(X = x)\,\mathbb{P}(Z = z)}\right)$$

$$= \sum_x \mathbb{P}(X = x) \sum_z \mathbb{P}(Z = z \mid X = x) \log \left( \frac{\mathbb{P}(Z = z \mid X = x))}{\mathbb{P}(Z = z)} \right).$$

Similarly, one can define the mutual information for continuous variables. In particular, we are interested in the case where $X$ is discrete and $Y$ is continuous, denote by $\mu_{\mathcal{Y}}$ the marginal of $(X, Y)$ over $Y$ and by $\mu|_x$ the conditional $(Y \mid X = x)$.

$$I(X, Y) = \sum_x \mathbb{P}(X = x) \int_y \mu|_x(dy) \log \left( \frac{\mu|_x(dy)}{\mu(dy)} \right).$$

Let us show the following version of the data processing inequality

$$I(X, \hat{X}(Y)) \leq I(X, Y). \tag{23}$$

To do so, we will use the conditional independence of $X$ and $\hat{X}$ given $Y$, which leads to

$$\mathbb{P}\left(X = x \mid \hat{X} = x'\right) = \int \mathbb{P}(X = x \mid Y = dy) \, \mathbb{P}\left(Y = dy \mid \hat{X} = z\right)$$

$$= \int \frac{\mathbb{P}(X = x)\mu|_x(dy)}{\mu(dy)} \, \mathbb{P}\left(Y = dy \mid \hat{X} = z\right).$$

Hence, using Jensen inequality,

$$I(X, \hat{X}) = H(X) - H\left(X \mid \hat{X}\right)$$

$$= H(X) + \sum_z \mathbb{P}(\hat{X} = z) \sum_x \mathbb{P}(X = x) \log(\mathbb{P}\left(X = x \mid \hat{X} = z\right))$$

$$= H(X) + \sum_z \mathbb{P}(\hat{X} = z) \sum_x \mathbb{P}(X = x) \log \left( \int \frac{\mathbb{P}(X = x)\mu|_x(dy)}{\mu(dy)} \, \mathbb{P}\left(Y = dy \mid \hat{X} = z\right) \right)$$

$$\leq H(X) + \sum_z \mathbb{P}(\hat{X} = z) \sum_x \mathbb{P}(X = x) \int \mathbb{P}\left(Y = dy \mid \hat{X} = z\right) \log \left( \frac{\mathbb{P}(X = x)\mu|_x(dy)}{\mu(dy)} \right)$$

$$= H(X) + \sum_x \mathbb{P}(X = x) \int \mu(dy) \log \left( \frac{\mathbb{P}(X = x)\mu|_x(dy)}{\mu(dy)} \right)$$

$$= \sum_x \mathbb{P}(X = x) \left( \int \mu(dy) \log \left( \frac{\mathbb{P}(X = x)\mu|_x(dy)}{\mu(dy)} \right) - \log(P(X = x)) \right)$$

$$= \sum_x \mathbb{P}(X = x) \int \mu(dy) \log \left( \frac{\mu|_x(dy)}{\mu(dy)} \right)$$

$$= I(X, Y).$$

We continue by computing the value of $I(X, Y)$, by definition and using Jensen inequality, we get

$$I(X, Y) = \frac{1}{N} \sum_{i \in [N]} \int_{\mathcal{D}_n \sim \rho_i^{\otimes n}} \rho_i^{\otimes n}(d\mathcal{D}_n) \log \left( \frac{\rho_i^{\otimes n}(d\mathcal{D}_n)}{\frac{1}{N} \sum_{j \in [N]} \rho_j^{\otimes n}(d\mathcal{D}_n)} \right)$$

$$\leq \frac{1}{N} \sum_{i \in [N]} \int_{\mathcal{D}_n \sim \rho_i^{\otimes n}} \rho_i^{\otimes n}(d\mathcal{D}_n) \frac{1}{N} \sum_{j \in [N]} \log \left( \frac{\rho_i^{\otimes n}(d\mathcal{D}_n)}{\rho_j^{\otimes n}(d\mathcal{D}_n)} \right) = \frac{1}{N^2} \sum_{i,j \in [N]} K\left( \rho_i^{\otimes n} \middle\| \rho_j^{\otimes n} \right).$$

We conclude from the fact that for $p$ and $q$ two distributions on a space $\mathcal{Z}$, we have

$$K\left( p^{\otimes n} \middle\| q^{\otimes n} \right) = \int_{\mathcal{Z}^n} -\log \left( \frac{dp^{\otimes n}(z_1, \cdots, z_n)}{dq^{\otimes n}(z_1, \cdots, z_n)} \right) q^{\otimes n}(dz_1, \cdots, dz_n)$$

$$= \int_{\mathcal{Z}^n} -\log \left( \frac{\prod_{i \leq n} dp(z_i)}{\prod_{i \leq n} dq(z_i)} \right) q^{\otimes n}(dz_1, \cdots, dz_n)$$

$$= \sum_{i \leq n} \int_{\mathcal{Z}^n} -\log \left( \frac{dp(z_i)}{dq(z_i)} \right) q^{\otimes n}(dz_1, \cdots, dz_n)$$

$$= \sum_{i \leq n} \int_{\mathcal{Z}} - \log \left( \frac{\mathrm{d}p(z_i)}{\mathrm{d}q(z_i)} \right) q(\mathrm{d}z_i) = nK \, (p \, || \, q) \, .$$

This explains the result. □

Let us assemble all the results proven thus far. In order to reduce our excess risk to a quadratic metric, we have assumed that the conditional distribution $\rho_i|_x$ to be Gaussian noise. In order to integrate this constraint into the precedent derivations, we leverage the following lemma.

**Lemma 13** (Kullback-Leibler divergence with Gaussian noise). *If $\rho_i$ and $\rho_j$ are two different distributions on $\mathcal{X} \times \mathcal{Y}$ such that there marginal over $\mathcal{X}$ are equal and the conditional distributions $(Y \, | \, X = x)$ are respectively equal to $\mathcal{N}(f_i(x), \sigma I_m)$ and $\mathcal{N}(f_j(x), \sigma I_m)$, then*

$$K \, (\rho_i \, || \, \rho_j) = \frac{1}{2\sigma^2} \left\| f_i - f_j \right\|^2_{L^2(\rho_{\mathcal{X}})} \, .$$

*Proof.* We proceed with

$$K \, (\rho_i \, || \, \rho_j) = \int_{\mathcal{X}} \mathbb{E}_{Y \sim \mathcal{N}(f_j(x), \sigma I_m)} \left[ \frac{\|Y - f_i(x)\|^2 - \|Y - f_j(x)\|^2}{2\sigma^2} \right] \rho_j(\mathrm{d}x)$$

$$= \int_{\mathcal{X}} \mathbb{E}_{Y \sim \mathcal{N}\left( \frac{f_j(x) - f_i(x)}{\sqrt{2}\sigma}, I_m \right)} \left[ \|Y\|^2 \right] - \mathbb{E}_{Y \sim \mathcal{N}(0, I_m)} \left[ \|Y\|^2 \right] \rho_j(\mathrm{d}x)$$

$$= \int_{\mathcal{X}} \left( m + \frac{\|f_j(x) - f_i(x)\|^2}{2\sigma^2} - m \right) \rho_j(\mathrm{d}x) = \frac{\|f_j - f_i\|^2_{L^2(\rho_{\mathcal{X}})}}{2\sigma^2},$$

where we have used the fact that the mean of a non-central $\chi$-square variable of parameter $(m, \mu^2)$ is $m + \mu^2$. One could also develop the first two squared norms and use the fact that for any vector $u \in \mathbb{R}^m$, $\mathbb{E}[\langle Y - f_i(x), u \rangle] = 0$ to get the result. □

Combining the different results leads to the following proposition.

**Lemma 14.** *Under Assumption 1 with $\mathcal{F} = \{x \in \mathcal{X} \to \theta\varphi(x) \in \mathcal{Y} \, | \, \|\theta\| \leq M\}$ and $\varphi$ bounded by $\kappa$, for any family $(f_i)_{i \leq N_\varepsilon} \in \mathcal{F}^N$ and any $\sigma > 2M\kappa\sigma_m$*

$$\inf_{\mathcal{A}} \sup_{\rho} \mathbb{E}_{\mathcal{D}_n \sim \rho^{\otimes n}} [\mathcal{R}(f_{\mathcal{A}(\mathcal{D}_n)}; \rho)] - \mathcal{R}^*(\rho)$$

$$\geq \frac{\min_{i,j \in [N]} \|f_i - f_j\|^2_{L^2(\rho_{\mathcal{X}})}}{16(m+2)^{1/2}\sigma} \left( 1 - \frac{\log(2)}{\log(N)} - \frac{n \max_{i,j \in [N]} \|f_i - f_j\|^2_{L^2(\rho_{\mathcal{X}})}}{2\sigma^2 \log(N)} \right),$$

*for any algorithm $\mathcal{A}$ that maps a dataset $\mathcal{D}_n \in (\mathcal{X} \times \mathcal{Y})^n$ to a parameter $\theta \in \Theta$.*

### A.3.2 Covering number for linear model

We are left with finding a good packing of the space induced by Assumption 1. To do so, we shall recall some property of reproducing kernel methods.

**Lemma 15** (Linear models are ellipsoids). *For $\mathcal{H}$ a separable Hilbert space and $\varphi : \mathcal{X} \to \mathcal{H}$ bounded, the class of functions $\mathcal{F} = \{x \in \mathcal{X} \to \theta\varphi(x) \in \mathcal{Y} \, | \, \|\theta\| \leq M\}$ can be characterized by*

$$\mathcal{F} = \left\{ f : \mathcal{X} \to \mathcal{Y} \, \middle| \, \left\| K^{-1/2} f \right\|_{L^2(\rho_{\mathcal{X}})} \leq M \right\}, \tag{24}$$

*where $\rho_{\mathcal{X}}$ is any distribution on $\mathcal{X}$ and $K$ is the operator on $L^2(\rho_{\mathcal{X}})$ that map $f$ to*

$$Kf(x') = \int_{x \in \mathcal{X}} \langle \varphi(x), \varphi(x') \rangle f(x) \rho_{\mathcal{X}}(\mathrm{d}x),$$

*whose image is assumed to be dense in $L^2$.*

*Proof.* This follows for isometry between elements in $\mathcal{H}$ and elements in $L^2$. More precisely, let us define

$$S: \quad \mathcal{Y} \otimes \mathcal{H} \quad \to \quad L^2(\mathcal{X}, \mathcal{Y}, \rho_\mathcal{X})$$
$$\theta \quad \to \quad x \to \theta\varphi(x).$$

The adjoint of $S$ is characterized by

$$S^*: \quad L^2(\mathcal{X}, \mathcal{Y}, \rho_\mathcal{X}) \quad \to \quad \mathcal{Y} \otimes \mathcal{H}$$
$$f \quad \to \quad \mathbb{E}[f(x) \otimes \varphi(X)],$$

which follows from the fact that for $\theta \in \mathcal{Y} \otimes \mathcal{H}$, $f \in L^2$ we have

$$\langle \theta, S^* f \rangle_{\mathcal{Y} \otimes \mathcal{H}} = \langle S\theta, f \rangle_{L^2} = \sum_{i=1}^{m} \int_\mathcal{X} f_i(x) \langle \theta_i, \varphi(x) \rangle_\mathcal{H} \, \rho_\mathcal{X}(\mathrm{d}x)$$

$$= \sum_{i=1}^{m} \langle \theta_i, \mathbb{E}[f_i(X)\varphi(X)] \rangle_\mathcal{H} = \langle \theta, \mathbb{E}[f(X) \otimes \varphi(X)] \rangle_{\mathcal{Y} \otimes \mathcal{H}} .$$

When $SS^*$ is compact and dense in $L^2$, we have

$$\|\theta\|_{\mathcal{Y} \otimes \mathcal{H}} = \left\| (SS^*)^{-1/2} S\theta \right\|_{L^2(\rho_\mathcal{X})} .$$

The compactness allows considering spectral decomposition hence fractional powers. We continue by observing that $SS^* = K$, which follows from

$$(SS^* f)(x') = (S \mathbb{E}[f(X) \otimes \varphi(X)])(x') = \mathbb{E}[f(X) \otimes \varphi(X)]\varphi(x') = \mathbb{E}[\langle \varphi(X), \varphi(x') \rangle f(X)].$$

The compactness of $K$ derives from the fact that

$$\|Kf(x')\|^2 = \|\mathbb{E}[\langle \varphi(X), \varphi(x') \rangle f(X)]\|^2 \le \mathbb{E}[\|\langle \varphi(X), \varphi(x') \rangle f(X)\|^2] \le \kappa^2 \|f\|_{L^2}^2 .$$

Hence, $\|K\|_{\mathrm{op}} \le \kappa^2$. Indeed, it is not hard to prove that the trace of $K$ is bounded by $m\kappa^2$, hence $K$ is not only compact but trace-class. $\square$

It should be noted that the condition on $K$ being dense in $L^2(\rho_\mathcal{X})$ is not restrictive, as indeed all the problem is only seen through the lens of $\varphi$ and $\rho_\mathcal{X}$: one can replace $\mathcal{X}$ by $\mathrm{supp}\,\rho_\mathcal{X}$ and $L^2(\rho_\mathcal{X})$ by the closure of the range of $K$ in $L^2(\rho_\mathcal{X})$ without modifying nor the analysis, nor the original problem.

We should study packing in the ellipsoid $\mathcal{F} = \left\{ f \in L^2 \,\middle|\, \|K^{-1/2}f\|_{L^2(\rho_\mathcal{X})} \le M \right\}$. It is useful to split the ellipsoid between a projection on a finite dimensional space that is isomorphic to the Euclidean space $\mathbb{R}^k$ and on a residual space $R$ where the energies $(\|f|_R\|_{L^2(\rho_\mathcal{X})})^2_{f \in \mathcal{F}}$ are uniformly small. We begin with the following packing lemma, sometimes referred to as Gilbert-Varshamov bound (Gilbert, 1952; Varshamov, 1957) which corresponds to a more generic result in coding theory.

**Lemma 16** ($\ell_2^2$-packing of the hypercube). *For any $k \in \mathbb{N}^*$, there exists a $k/4$-packing of the hypercube $\{0,1\}^k$, with respect to Hamming distance, of cardinality $N = \exp(k/8)$.*

*Proof.* Let us consider $\varepsilon > 0$, and a maximal $\varepsilon$-packing $(x_i)_{i \le N}$ of the hypercube with respect to the distance $d(x, y) = \sum_{i \in [k]} \mathbf{1}_{x_i \ne y_i} = \|x - y\|_1 = \|x - y\|_2^2$. By maximality, we have $\{0,1\}^k \subset \cup_{i \in [N]} B_d(x_i, \varepsilon)$, hence

$$2^k \le N \left| \{x \in \{0,1\}^k \,|\, \|x\|_1 \le \varepsilon\} \right|.$$

This inequality can be rewritten with $Z$ a binomial variable of parameter $(k, {}^1\!/\!{}_2)$ as $1 \le N \mathbb{P}(Z \le \varepsilon)$. Using Hoeffding inequality (Hoeffding, 1963), when $\varepsilon = k/4$ we get

$$N^{-1} \le \mathbb{P}(Z \le k/4) = \mathbb{P}(Z - \mathbb{E}[Z] \le k/4) \le \exp\left(-\frac{2k^2}{4^2 k}\right) = \exp\left(-k/8\right) .$$

This is the desired result. $\square$

**Lemma 17** (Packing of infinite-dimensional ellipsoids). *Let $\mathcal{F}$ be the function in $L^2(\rho_\mathcal{X})$ such that $\left\|K^{-1/2}f\right\|_{L^2(\rho_\mathcal{X})} \le M$ for $K$ a compact operator and $M$ any positive number. For any $k \in \mathbb{N}^*$, it is possible to find a family of $N \ge \exp(k/8)$ elements $(f_i)_{i \in [N]}$ in $\mathcal{F}$ such that for any $i \ne j$,*

$$\frac{kM^2}{\sum_{i \le k} \lambda_i^{-1}} \le \|f_i - f_j\|_{L^2(\rho_\mathcal{X})}^2 \le \frac{4kM^2}{\sum_{i \le k} \lambda_i^{-1}}, \tag{25}$$

*where $(\lambda_i)_{i \in \mathbb{N}}$ are the ordered (with repetition) eigenvalues of $K$.*

*Proof.* Let us denote by $(\lambda_i)_{i \in \mathbb{N}}$ the eigenvalues of $K$ and $(u_i)_{i \in \mathbb{N}}$ in $L^2$ the associated eigenvectors. Consider $(a_s)_{s \in [N]}$ a $k$-packing of the hypercube $\{-1, 1\}^k$ for $N \geq \exp(k/8)$ with respect to the $\ell_2^2$ quasi-distance and define for any $a \in \{a_s\}$

$$f_a = \frac{M}{c} \sum_{s=1}^{k} a_i u_i,$$

with $c^2 = \sum_{i=1}^{k} \lambda_i^{-1}$. We verify that

$$\left\| K^{-1/2} f_a \right\|_{L^2}^2 = \frac{M^2}{c^2} \sum_{i=1}^{k} \lambda_i^{-1} = M^2.$$

$$\|f_a - f_b\|_{L^2}^2 = \frac{M^2}{c^2} \sum_{i=1}^{k} |a_i - b_i|^2 = \frac{M^2}{c^2} \|a_i - b_i\|_2^2 \in \frac{M^2}{c^2} \cdot [k, 4k].$$

This is the object of the lemma. $\qquad\square$

So far, we have proven the following lower bound.

**Lemma 18.** *Under Assumption 1 with $\mathcal{F} = \{x \in \mathcal{X} \to \theta\varphi(x) \in \mathcal{Y} \mid \|\theta\| \leq M\}$ and $\varphi$ bounded by $\kappa$, for any family $(f_i)_{i \leq N_\varepsilon} \in \mathcal{F}^N$ and any $\sigma > 2M\kappa\sigma_m$ and $km > 10$,*

$$\inf_{\mathcal{A}} \sup_{\rho} \mathbb{E}_{\mathcal{D}_n \sim \rho^{\otimes n}} [\mathcal{R}(f_{\mathcal{A}(\mathcal{D}_n)}; \rho)] - \mathcal{R}^*(\rho) \geq \frac{1}{128} \min \left\{ \frac{M^2}{\sigma m^{1/2} \sum_{i \leq k} (k\lambda_i)^{-1}}, \frac{\sigma k m^{1/2}}{32n} \right\},$$

*for any algorithm $\mathcal{A}$ that maps a dataset $\mathcal{D}_n \in (\mathcal{X} \times \mathcal{Y})^n$ to a parameter $\theta \in \Theta$, and where $(\lambda_i)$ are the ordered eigenvalue of the operator $K$ on $L^2(\mathcal{X}, \mathbb{R}, \rho_{\mathcal{X}})$ that maps any function $f$ to the function $Kf$ defines for $x' \in \mathcal{X}$ as*

$$(Kf)(x') = \int_{x \in \mathcal{X}} \langle \varphi(x), \varphi(x') \rangle f(x) \rho_{\mathcal{X}}(\mathrm{d}x).$$

*In particular, when $\lambda_i = \kappa^2 i^{-a}/\zeta(\alpha)$, where $\zeta$ denotes the Riemann zeta function, we get the following bounds. If we optimize with respect to $\sigma$, there exists $n_\alpha \in \mathbb{N}$ such that for any $n > n_\alpha$.*

$$\inf_{\mathcal{A}} \sup_{\rho} \mathbb{E}_{\mathcal{D}_n \sim \rho^{\otimes n}} [\mathcal{R}(f_{\mathcal{A}(\mathcal{D}_n)}; \rho)] - \mathcal{R}^*(\rho) \geq \frac{M\kappa}{725\zeta(\alpha)^{1/2} n^{1/2}}. \tag{26}$$

*If we fix $\sigma = \beta M\kappa$ with $\beta \geq 2$, and we optimize with respect to $k$, there exists a constant $c_\beta$ and an integer $n_0$ such that for $n > n_0$ we have*

$$\inf_{\mathcal{A}} \sup_{\rho} \mathbb{E}_{\mathcal{D}_n \sim \rho^{\otimes n}} [\mathcal{R}(f_{\mathcal{A}(\mathcal{D}_n)}; \rho)] - \mathcal{R}^*(\rho) \geq \frac{M\kappa c_\beta}{\zeta(\alpha)^{\frac{1}{1+\alpha}} n^{\frac{\alpha}{\alpha+1}}}. \tag{27}$$

*Proof.* Reusing Lemma 14, with the same notations, we have the lower bound in

$$\frac{\min \|f_i - f_j\|^2}{16\sigma(m+2)^{1/2}} \left( 1 - \frac{\log(2)}{\log(N)} - \frac{n \max \|f_i - f_j\|^2}{2\sigma^2 \log(N)} \right).$$

Let $K$ and $K_{\mathcal{Y}}$ be the self-adjoint operators on $L^2(\mathcal{X}, \mathbb{R}, \rho_{\mathcal{X}})$ and $L^2(\mathcal{X}, \mathcal{Y}, \rho_{\mathcal{X}})$ respectively, both defined through the formula

$$(Kf)(x') = \int_{x \in \mathcal{X}} \langle \varphi(x), \varphi(x') \rangle f(x) \rho_{\mathcal{X}}(\mathrm{d}x).$$

When $K$ is compact, it admits an eigenvalue decomposition $K = \sum_{i \in \mathbb{N}} \lambda_i u_i \otimes u_i$ where the equality as to be understood as the convergence of operator with respect to the operator norm based on the $L^2$-topology. It follows from the product structure of $L^2(\mathcal{X}, \mathcal{Y}, \rho_{\mathcal{X}}) \simeq L^2(\mathcal{X}, \mathbb{R}, \rho_{\mathcal{X}})^m$ that $K_{\mathcal{Y}} = \sum_{i \in \mathbb{R}, j \in [m]} \sum_{i \in \mathbb{N}, j \in [m]} \lambda_i (e_i \otimes y_j) \otimes (e_i \otimes u_j)$ with $(e_j)$ the canonical basis of $\mathcal{Y} = \mathbb{R}^m$. As a

consequence, if $(\lambda_i)_{i \in \mathbb{N}}$ are the ordered eigenvalues of $K$ then $(\lambda_{\lfloor i/m \rfloor})$ are the ordered eigenvalues of $K_{\mathcal{Y}}$. Hence, with Lemmas 15 and 17, it is possible to find $N = \exp(km/8)$ functions in $\mathcal{F}$ such that

$$\frac{kmM^2}{m \sum_{i \le k} \lambda_i^{-1}} \le \left\| f_i - f_j \right\|_{L^2(\rho_{\mathcal{X}})}^2 \le \frac{4kmM^2}{m \sum_{i \le k} \lambda_i^{-1}}.$$

If we multiply those functions by $\eta \in [0, 1]$ we get a lower bound in

$$\frac{\eta^2 M^2}{16\sigma(m+2)^{1/2} \sum_{i \le k} (k\lambda_i)^{-1}} \left( 1 - \frac{8 \log(2)}{km} - \frac{16M^2 n \eta^2}{\sigma^2 km \sum_{i \le k} (k\lambda_i)^{-1}} \right).$$

Making sure that the last two terms are smaller than one fourth and one half respectively we get the following conditions on $k$ and $\eta$, with $\Lambda_k = \sum_{i \le k} (k\lambda_i)^{-1}$,

$$km \ge 32 \log(2), \qquad 32 M^2 n \eta^2 \le \sigma^2 km \Lambda_k.$$

Using the fact that $\eta < 1$, the lower bound becomes

$$\frac{M^2}{128 \sigma m^{1/2} \Lambda_k} \min \left\{ 1, \frac{\sigma^2 km\Lambda_k}{32M^2 n} \right\} = \frac{1}{128} \min \left\{ \frac{M^2}{\sigma m^{1/2} \Lambda_k}, \frac{\sigma km^{1/2}}{32n} \right\},$$

as long as $km > 10$. When $\lambda_i^{-1} = i^\alpha \zeta(\alpha)/\kappa^2$, since $\Lambda_k \le \lambda_k^{-1}$, we simplify the last expression as

$$\frac{1}{128} \min \left\{ \frac{M^2 \kappa^2}{\sigma m^{1/2} k^\alpha \zeta(\alpha)}, \frac{\sigma km^{1/2}}{32n} \right\}.$$

Optimizing with respect to $\sigma$ leads to

$$\sigma^2 = \frac{32n M^2 \kappa^2}{mk^{1+\alpha} \zeta(\alpha)} \ge 4M^2 \kappa^2 \sigma_m.$$

This gives

$$n_{\alpha,m} = m\zeta(\alpha)\sigma_m^2/8.$$

The dependency of $n_\alpha$ to $m$ can be removed since any problem with $\mathcal{Y} = \mathbb{R}^m$ can be cast as a problem in $\mathbb{R}^{m+1}$ by adding a spurious coordinate. Taking $k = 1$ and $m = 10$ leads to the result stated in the lemma. When $n < n_\alpha$, one can artificially multiply the bound by $n_\alpha^{1/2}$, since an optimal algorithm can not do better with fewer data. After checking that one can take $\sigma_1 \ge 1$, this leads to a bound in

$$\frac{M\kappa}{2048 n^{1/2}}.$$

Optimizing with respect to $k$ leads to $k^{\alpha+1} = 32M^2 \kappa^2 n/(\sigma^2 m \zeta(\alpha))$ and a bound in

$$\frac{(\sigma m^{1/2})^{\frac{\alpha-1}{\alpha+1}} (M\kappa)^{\frac{2}{\alpha+1}}}{128(32n)^{\frac{\alpha}{\alpha+1}} \zeta(\alpha)^{\frac{1}{\alpha+1}}}.$$

The condition $k > \min \left\{ 10m^{-1}, 1 \right\}$ and $\sigma \ge 2M\kappa\sigma_m$ translates into the condition

$$4M^2 \kappa^2 \sigma_m^2 \le \sigma^2 \le \frac{32M^2 \kappa^2 n}{m\zeta(\alpha)} \min \left\{ 1, \frac{m^{1+\alpha}}{10^{1+\alpha}} \right\}.$$

Once again we can remove the dependency to $m$. Considering $\sigma = \beta M\kappa$ leads to the result stated in the lemma. $\square$

### A.3.3 Controlling eigenvalues decay

Based on Lemma 18, in order to prove Theorem 2, we only need to show that there exists a mapping $\varphi$, an input space $\mathcal{X}$ and a distribution $\rho_{\mathcal{X}}$ such that the integral operator $K$ introduced in the lemma verifies the assumption on its eigenvalues. Notice that we show in the proof of Lemma 18 that the universal constant $c_3$ can be taken as $c_3 = 2^{-11}$.

To proceed, let us consider any infinite dimensional Hilbert space $\mathcal{H}$ with a basis $(e_i)_{i \in \mathbb{N}}$, $\mathcal{X} = \mathbb{N}$ and $\varphi : \mathbb{N} \to \mathcal{H}; i \to \kappa e_i$. For $a : \mathbb{N} \to \mathbb{R}$ we have

$$(Ka)(i) = \sum_{j \in \mathbb{N}} \langle \varphi(i), \varphi(j) \rangle a(j)\rho(j) = \kappa^2 a(i)\rho(i).$$

Hence, the eigenvalues of $K$ are $(\kappa^2 \rho_{\mathcal{X}}(i))_{i \leq n}$. It suffices to consider $\rho_{\mathcal{X}}(i) = i^{-\alpha}/\zeta(\alpha)$ to conclude.

The eigenvalue decay in $O(i^{-\alpha})$ can also be witnessed in many regression problems. One way to build those cases is to turn a sequence of non-negative real values into a one-periodic function $h$ from $\mathbb{R}^d$ to $\mathbb{R}$ thanks to the inverse Fourier transform. Using Bochner (1933), one can construct a map $\varphi$ such that the convolution operator linked with $h$ corresponds to the operator $K$. When $\rho$ is uniform on $[0,1]^d$, diagonalizing this convolution operator with the Fourier functions and using the property in Lemma 15 shows that the class of functions $\mathcal{F}$ are akin to Sobolev spaces. Similar behavior can be proven when $\mathcal{X} = \mathbb{R}^d$ and $\rho_{\mathcal{X}}$ is absolutely continuous with respect to the Lebesgue measure and has bounded density (Widom, 1963). We refer the curious reader to Scholkopf and Smola (2001) or Bach (2023) for details.

# B  Unbiased weakly supervised stochastic gradients

In this section, we provide a generic scheme to acquire unbiased weakly supervised stochastic gradients, as well as specifications of the formula given in the main text for least-squares and median regression.

## B.1  Generic implementation

Suppose that $\Theta$ is finite dimensional, or that it can be approximated by a finite dimensional space without too much approximation error. For example, in the realm of scalar-valued kernel methods, it is usual to consider either the random finite dimensional space $\text{Span} \{\varphi(x_i)\}_{i \leq n}$ for $(x_i)$ the data points, or the finite dimension space linked to the first eigenspaces of the operator $\mathbb{E}[\varphi(X) \otimes \varphi(X)]$. In the context of neural networks, the parameter space is always finite-dimensional.

Suppose also that, given $\theta$, we know an upper bound $M_\theta$ on the amplitude of $\nabla_\theta \ell(f_\theta(x), y)$, or that we know how to handle clipped gradients at amplitude $M_\theta$ for SGD. Then, similarly to the least-squares method proposed in the main text, we can access weakly supervised gradient through the formula

$$\nabla_\theta \ell(f_\theta(x), y) = \frac{2M_\theta(|\Theta|^2 + 4|\Theta| + 3)}{\pi^{3/2}} \mathbb{E}_{U \sim \mathcal{U}(B_\Theta), V \sim \mathcal{U}([0, M_\theta])} [\mathbf{1}_{y \in (z \to \langle U, \nabla_\theta \ell(f_\theta(x), z) \rangle)^{-1}([V, \infty))} U],$$

where $B_\Theta$ is the unit ball of $\Theta$.

This scheme is really generic, and we do not advocate for it in practice as one may hope to leverage specific structure of the loss function and the parametric model in a more efficient way. This formula is rather a proof of concept to illustrate that our technique can be applied generically, and is not specific to least-squares or median regression.

## B.2  Specific implementations

Let us prove the two formulas to get stochastic gradients for both least-squares and median regression. We begin with median regression. Consider $z \in \mathbb{S}^{m-1}$, and let us denote

$$x = \mathbb{E}_U[\text{sign}(\langle z, U \rangle) U].$$

The direction $x/\|x\| \in \mathbb{S}^{m-1}$ is characterized by the argmax over the sphere of the linear form

$$y \to \langle \mathbb{E}_U[\text{sign}(\langle z, U \rangle) U], y \rangle = \mathbb{E}_U[\text{sign}(\langle z, U \rangle) \langle U, y \rangle].$$

This linear form has a unique maximizer on $\mathbb{S}^{m-1}$ and by invariance by symmetry over the axis $z$, this maximizer is aligned with $z$, hence $x = c_x \cdot z$. We compute the amplitude with the formula, because $z$ is a unit vector

$$c_x = \langle x, z \rangle = \mathbb{E}_U[\text{sign}(\langle z, U \rangle) \langle U, z \rangle].$$

By invariance by rotation of both the uniform distribution and the scalar product, $c_x$ is actually a constant, it is equal to its value $c_2 = c_{e_1}$.

The same type of reasoning applies for the least-squares case. Consider $z \in \mathbb{R}^m$, and denote

$$x = \mathbb{E}_{U,V} [\mathbf{1}_{\langle z, U \rangle \geq V} \cdot U].$$

For the same reasons as before $x = c_x \cdot u$ for $u = z/\|z\|$, and $c_x$ verifies

$$c_x = \langle x, u \rangle = \mathbb{E}_{U,V}[\mathbf{1}_{\langle z, U \rangle \geq V} \langle U, u \rangle] = \mathbb{E}_U[\mathbb{E}_V[\mathbf{1}_{\langle z, U \rangle \geq V}] \langle U, u \rangle]$$

$$= \mathbb{E}_U [\mathbf{1}_{\langle z,U\rangle>0} \frac{\langle z, U\rangle}{M} \langle U, u\rangle] = \frac{\|z\|}{M} \mathbb{E}_U [\mathbf{1}_{\langle u,U\rangle>0} \langle U, u\rangle^2].$$

Hence,

$$x = \frac{1}{M} \mathbb{E}_U [\mathbf{1}_{\langle u,U\rangle>0} \langle U, u\rangle^2] \cdot z = c_1 \cdot z.$$

This explains the formula for least-squares.

**Lemma 19** (Constant for the uniform strategy). *Under the uniform distribution on the sphere*

$$c_2 = \mathbb{E}_{u\sim\mathbb{S}^{m-1}} [|\langle u, e_1\rangle|] = \frac{\sqrt{\pi}\,\Gamma(\frac{m-1}{2})}{m\,\Gamma(\frac{m}{2})} \geq \frac{\sqrt{2\pi}}{m^{3/2}}. \tag{28}$$

*Proof.* Let us compute $c_2 = \mathbb{E}_{u\sim\mathbb{S}^{m-1}} [|\langle u, e_1\rangle|]$. This constant can be written explicitly as

$$c_2 = \frac{\int_{x\in\mathbb{S}^{m-1}} |x_1|\,\mathrm{d}x}{\int_{x\in\mathbb{S}^{m-1}} \mathrm{d}x}.$$

Remark that for any function $f : \mathbb{R} \to \mathbb{R}$, we have

$$\int_{\mathbb{S}^{m-1}} f(x_1)\,\mathrm{d}x = \int_{x_1\in[-1,1]} f(x_1)\,\mathrm{d}x_1 \int_{\tilde{x}\in\sqrt{1-x_1^2}\cdot\mathbb{S}^{m-2}} \mathrm{d}\tilde{x} = \int_{x_1\in[-1,1]} f(x_1)(1 - x_1^2)^{\frac{m-2}{2}}\,\mathrm{d}x_1 \int_{\tilde{x}\in\mathbb{S}^{m-2}} \mathrm{d}\tilde{x}.$$

By denoting $S_m$ the surface of the $m$-sphere, the last integral is nothing but $S_{m-2}$. By setting $f(x) = 1$, we can retrieve by recurrence the expression of $S_m$. In our case, $f(x) = |x|$, so we compute, with $u = 1 - x^2$

$$\int_{x_1\in[-1,1]} |x_1|\,(1 - x_1^2)^{\frac{m-2}{2}}\,\mathrm{d}x_1 = 2\int_{x_1\in[0,1]} x_1(1 - x_1^2)^{\frac{m-2}{2}}\,\mathrm{d}x_1 = \int_{u=0}^1 u^{\frac{m-2}{2}}\,\mathrm{d}u = \frac{1}{m}.$$

This leads to

$$c_2 = \frac{S_{m-2}}{mS_{m-1}} = \frac{\sqrt{\pi}\,\Gamma(\frac{m-1}{2})}{m\,\Gamma(\frac{m}{2})}.$$

The ratio $S_{m-2}/S_{m-1}$ can be expressed with the integral corresponding to $f = 1$, but it is common knowledge that $S_{m-1} = 2\pi^{m/2}/\Gamma(m/2)$. $\square$

**Lemma 20** (Constant for least-squares). *Under the uniform distributions on $[0, M]$ and the sphere*

$$c_1 = \mathbb{E}_{y\sim[0,M]} \mathbb{E}_{u\sim\mathbb{S}^{m-1}} [\mathbf{1}_{\langle u,e_1\rangle>v} \langle u, e_1\rangle] = \frac{\pi^{3/2}}{M(m^2 + 4m + 3)}. \tag{29}$$

*Proof.* Similarly to the previous case, this constant can be written explicitly as

$$c_1 = \frac{1}{2} \frac{\int_{y\in[0,M]} \int_{x\in\mathbb{S}^{m-1}} |x_1|\,\mathbf{1}_{|x_1|>y}\,\mathrm{d}y\,\mathrm{d}x}{M\int_{x\in\mathbb{S}^{m-1}} \mathrm{d}x} = \frac{\int_{x\in\mathbb{S}^{m-1}} x_1^2\,\mathrm{d}x}{2M\int_{x\in\mathbb{S}^{m-1}} \mathrm{d}x}.$$

We continue as before with

$$\int_{x_1\in[-1,1]} |x_1|^2\,(1 - x_1^2)^{\frac{m-2}{2}}\,\mathrm{d}x_1 = 2\int_{x\in[0,1]} x^2(1 - x^2)^{\frac{m-2}{2}}\,\mathrm{d}x = \frac{2\pi\Gamma(\frac{m}{2})}{4\Gamma(\frac{m+3}{2})}.$$

This leads to

$$c_1 = \frac{\pi\Gamma(\frac{m}{2})}{4M\Gamma(\frac{m+3}{2})} \cdot \frac{\sqrt{\pi}\,\Gamma(\frac{m-1}{2})}{\Gamma(\frac{m}{2})} = \frac{\pi^{3/2}\Gamma(\frac{m-1}{2})}{4M\Gamma(\frac{m+3}{2})} = \frac{\pi^{3/2}}{M(m^2 + 4m + 3)}.$$

This is the result stated in the lemma. $\square$

## C  Median surrogate

Let us begin this section by proving Proposition 3. This result is actually the integration over $x \in \mathcal{X}$ of a pointwise result, so let us fix $x \in \mathcal{X}$. Consider a probability distribution $p \in \Delta_\mathcal{Y}$ over $\mathcal{Y}$, and its median $\Theta^* \subset \mathbb{R}^\mathcal{Y}$ defined as the minimizer of $\mathcal{R}_S(\theta) = \mathbb{E}_p[\|\theta - e_Y\|]$. We will to prove that $\cup_{\theta \in \Theta^*} \arg\max_{y \in \mathcal{Y}} \theta_y = \arg\max_{y \in \mathcal{Y}} p(y)$.

Let us begin by the inclusion $\arg\max_{y \in \mathcal{Y}} p(y) \subset \cup_{\theta \in \Theta^*} \arg\max_{y \in \mathcal{Y}} \theta_y$. To do so, consider $\theta \in \mathbb{R}^\mathcal{Y}$ and $\sigma \in \mathfrak{S}_\mathcal{Y}$ the transposition of two elements $y$ and $z$ in $\mathcal{Y}$. Denote by $\theta_\sigma \in \mathbb{R}^\mathcal{Y}$, the vector such that $(\theta_\sigma)_{y'} = \theta_{\sigma(y')}$ for any $y' \in \mathcal{Y}$. We have

$$
\begin{aligned}
\mathcal{R}_S(\theta) - \mathcal{R}_S(\theta_\sigma) &= \sum_{y' \in \mathcal{Y}} p(y') \left( \|\theta - e_{y'}\| - \|\theta_\sigma - e_{y'}\| \right) \\
&= \sum_{y' \in \mathcal{Y}} p(y') \left( \sqrt{\sum_{z' \in \mathcal{Y}} \theta_{z'}^2 + (1 - \theta_{y'})^2 - \theta_{y'}^2} - \sqrt{\sum_{z' \in \mathcal{Y}} \theta_{\sigma(z')}^2 + (1 - \theta_{\sigma(y')})^2 - \theta_{\sigma(y')}^2} \right) \\
&= (p(y) - p(z)) \left( \sqrt{\sum_{z' \in \mathcal{Y}} \theta_{z'}^2 + 1 - 2\theta_y} - \sqrt{\sum_{z' \in \mathcal{Y}} \theta_{z'}^2 + 1 - 2\theta_z} \right).
\end{aligned}
$$

Because, for any $a \in \mathbb{R}_+$, the function $x \to \sqrt{a - 2x}$ is increasing, if $p(y) > p(z)$, then to minimize $\mathcal{R}$, we should make sure that $\theta_y \geq \theta_z$ i. As a consequence, because of symmetry, the modes of $p$ do correspond to argmax of $(\theta_y^*) y \in \mathcal{Y}$ for some $\theta^* \in \Theta^*$.

Let us now prove the second inclusion. To do so, suppose that $p(1) > p(2)$, and let us show that $\theta_1^* > \theta_2^*$. Let us parametrize $\theta_1 = a + \varepsilon$ and $\theta_2 = a - \varepsilon$ for a given $a$, and show that $\varepsilon = 0$ is not optimal in order to minimize the risk $\mathcal{R}_S$ seen as a function of $\varepsilon$. To do so, we can use the Taylor expansion of $\sqrt{1 + x} = 1 + x/2$. Hence, with $A = \sum_{y>2} (\theta_y^*)^2$, retaking the last derivations

$$
\begin{aligned}
\mathcal{R}_S(\varepsilon) &= p(1) \sqrt{(a + \varepsilon)^2 + (a - \varepsilon)^2 + A + 1 - 2(a + \varepsilon)} \\
&\quad + p(2) \sqrt{(a + \varepsilon)^2 + (a - \varepsilon)^2 + A + 1 - 2(a - \varepsilon)} \\
&\quad + \sum_{y>2} p(y) \sqrt{(a + \varepsilon)^2 + (a - \varepsilon)^2 + A + 1 - 2\theta_y^*} \\
&= p(1) \sqrt{2a^2 + 2\varepsilon^2 + A + 1 - 2a - 2\varepsilon} \\
&\quad + p(2) \sqrt{2a^2 + 2\varepsilon^2 + A + 1 - 2a + 2\varepsilon} + c + o(\varepsilon) \\
&= \tilde{c} + \frac{\varepsilon}{\sqrt{2a^2 + A + 1 - 2a}} (p(2) - p(1)) + o(\varepsilon).
\end{aligned}
$$

This shows that taking $\theta_1^* = \theta_2^*$, that is $\varepsilon = 0$, is not optimal, hence we have the second inclusion, which ends the proof. Note that we have proven a much stronger result, we have shown that $(\theta_y)$ and $p(y)$ are order in the exact same fashion (with respect to the strict comparison $p(y) > p(z) \Rightarrow \theta_y^* > \theta_z^*$ for any $\theta^* \in \Theta^*$).

### C.1  Discussion around the median surrogate.

The median surrogate have some nice properties for a surrogate method, in particular it does not fully characterize the distribution $p(y)$ in the sense that there is no one-to-one mapping from $p$ to $\theta^*$. For example, when $\mathcal{Y} = \{1, 2, 3\}$ if $p(y = e_1), p(y = e_2), p(y = e_3) \propto (1, 1, 2\cos(\pi/6))$, then the geometric median correspond to $\theta^* = e_3$. This differs from smooth surrogates, such as logistic regression or least-squares, that implicitly learn the full distribution $p$, which should be seen as a waste of resources. Non-smooth surrogates tend to exhibit faster rates of convergence (in terms of decrease of the original risk as a function of the number of samples) than smooth surrogates when rates are derived through calibration inequalities (Nowak-Vila, 2021). It would be nice to derive generic calibration inequality for the median surrogate for multiclass, and see how to derive a median surrogate for more structured problems such as ranking problems.

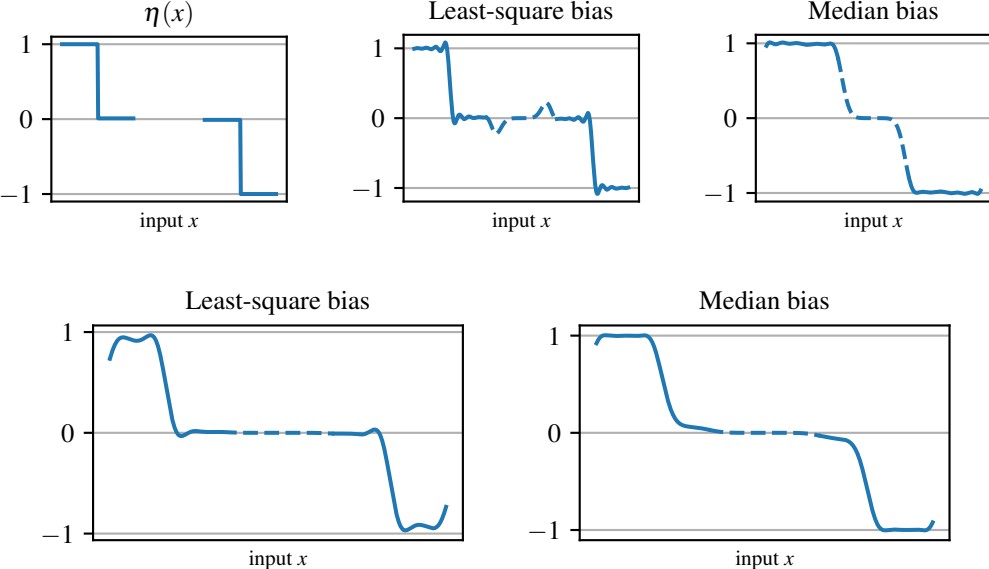

**Figure 3:** *Comparison of least-squares and absolute deviation with noise irregularity* for a classification problem specified by $\mathcal{X} = [0, 3]$, $\mathcal{Y} = \{-1, 1\}$ with $X$ uniform on $[0, 1] \cup [2, 3]$ and $\eta(x) = \mathbb{E}\{Y \mid X = x\}$ specified on the left figure. The optimal classifier, with respect to the zero-one loss, $f^*(x) = \text{sign}\, \eta$ takes value one on $[0, 1]$ and value minus one on $[2, 3]$. The regularized solution are defined as $\arg\min_g \mathbb{E}[\|\langle\varphi(X), \theta\rangle - Y\|^p] + \lambda \|\theta\|$ with $p = 2$ for least-squares (middle), and $p = 1$ for the median (right). They can be translated into classifiers with the decoding $f = \text{sign}\, g$. In this figure, we choose $\varphi$ implicitly through the Gaussian kernel $k(x, x') = \langle\varphi(x), \varphi(x')\rangle = \exp(-\|x - x'\|^2/2\sigma^2)$ with $\sigma = .1$ which explains the frequency of the observed oscillations, and choose $\lambda = 10^{-6}$ (top) and $\lambda = 10^{-2}$ (bottom). On the one hand, because the least-squares surrogate is trying to estimate $\eta$ it suffers from its lack of regularity, leading to Gibbs phenomena that restricts it to be a perfect classifier. On the other hand, the absolute deviation is trying to approach the function $f^*$ itself, and does not suffer from its lack of regularity. In this setting, if we approach the original classification problem by minimization of the surrogate empirical risks, and denote by $g_n$ this minimizer and $f_n = \text{sign}\, g_n$ its decoding, $f_n$ obtained through median regression will converge exponentially fast toward $f^*$, while $f_n$ obtained through least-squares will never converge to the solution $f^*$.

# D   Classification with a min-max game

In this section, we prove and extend on Proposition 4. First of all, let us consider the average loss, for $(v_y) \in \mathbb{R}^{\mathcal{Y}}$ summing to one

$$\bar{L}(v, s) = 1 - \sum_{y \in s} v_y = \sum_{y \notin s} v_y.$$

Consider now this loss conditioned on the observation $\mathbf{1}_{y \in s}$, we have plenty of characterizations of $L$,

$$L(v, s; \mathbf{1}_{y \in s} - \mathbf{1}_{y \notin s}) = \mathbf{1}_{y \in s} \bar{L}(v, s) + \mathbf{1}_{y \notin s} \bar{L}(v, \mathcal{Y} \setminus s) = \mathbf{1}_{y \in s} \sum_{y \notin s} v_y + \mathbf{1}_{y \notin s} \sum_{y \in S} v_y$$

$$= \mathbf{1}_{y \in s} + (\mathbf{1}_{y \notin s} - \mathbf{1}_{y \in s}) \sum_{y \in s} v_y = \mathbf{1}_{y \notin s} + (\mathbf{1}_{y \in s} - \mathbf{1}_{y \notin s}) \sum_{y \notin s} v_y$$

$$= \frac{1}{2} - \frac{1}{2}(\mathbf{1}_{y \in s} - \mathbf{1}_{y \notin s})\left(\sum_{y \in s} v_y - \sum_{y \notin s} v_y\right) = \frac{1}{2} + \frac{1}{2}(\mathbf{1}_{y \in s} - \mathbf{1}_{y \notin s})\left(1 - 2\sum_{y \in s} v_y\right).$$

Minimizing this loss or the loss $2L - 1$ as defined in Proposition 4 is equivalent.

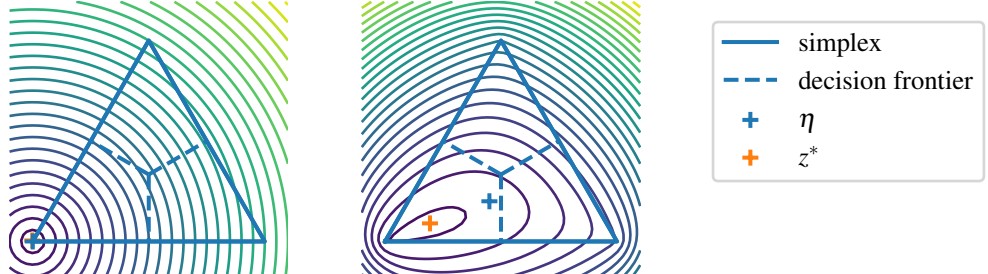

**Figure 4:** *Comparison of least-squares and median surrogate without context.* Consider a context-free classification problem that consists in estimating the mode of a distribution $p \in \Delta_{\mathcal{Y}}$, or equivalently the minimizer of the 0-1 loss. Such a problem can be visualized on the simplex $\Delta_m$ where $\mathcal{Y} = \{y_1, \cdots, y_m\} \simeq \{1, \cdots, m\}$ is mapped to the canonical basis $\{e_i\}_{i \in [m]} \in \mathbb{R}^m$. The figure illustrates the case $m = 3$. The least-squares and median surrogate methods can be understood as working in this simplex, estimating a quantity $z \in \Delta_{\mathcal{Y}}$, before performing the decoding $y(z) = \arg\max_y \langle z, e_y \rangle$. Such a decoding partitions the simplex in regions whose frontiers are represented in dashed blue on the figure. The distribution $p$ is characterized on the simplex by $\eta = \mathbb{E}_{Y \sim p}[e_Y] = \arg\min \mathbb{E}_{Y \sim p}[\|z - e_Y\|^2]$. This quantity $\eta$ is exactly the quantity estimated by the least-squares surrogate. The median surrogate searches the minimizer $z^*$ of the quantity $\mathcal{E}(z) = \mathbb{E}_{Y \sim p}[\|z - e_Y\|]$, whose level lines are represented in solid on the figure. One of the main advantage of the median surrogate compared to the least-squares one is that $z^*$ is always farther away from the boundary frontier than $\eta$, meaning that for a similar estimation error on this quantity, the error on the decoding, which corresponds to an estimate of the mode of $p$, will be much smaller for the median surrogate. The left figure represents the case $p = (1, 0, 0)$, the right figure the case $p = (.45, .35, .2)$.

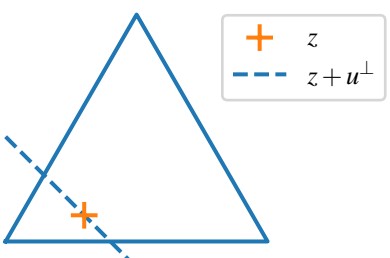

**Figure 5:** *Query strategy based on regression surrogate.* Retaking the simplex representation of Figure 4, the query strategy for classification approached with least-squares surrogate or median surrogate consists in looking at the current surrogate estimate $z$ in the simplex $\Delta_{\mathcal{Y}}$, taking a random direction $u \in \mathbb{R}^{\mathcal{Y}}$ and querying $\text{sign}(\langle e_Y - z, u \rangle)$. We see that with three elements, when $Y$ is deterministic, the optimal query strategy consists in considering $s = \{y\}$, while surrogate strategies, such as least-squares and median regression, that learn $z^* = e_y$, would only make such a query only two third of the time (which is the ratio of the solid angle of $[e_2, e_3]$ from $e_1$ divided by $\pi$). This shows that those surrogate strategies do not fully leverage the specific structure of the output.

## D.1 Consistency

Let us consider the loss as defined in this proposition, we have the characterization

$$L(v, s; \mathbf{1}_{y \in s} - \mathbf{1}_{y \notin s}) = (\mathbf{1}_{y \in s} - \mathbf{1}_{y \notin s}) \left( \sum_{y \in s} v_y - \sum_{y \notin s} v_y \right).$$

Let us rewrite (8) based on this previous characterization of the loss, we have

$$\mathbb{E}_Y[L(v, s, \mathbf{1}_{Y \in s} - \mathbf{1}_{Y \notin s})] = -(\mathbb{P}_Y(Y \in s) - \mathbb{P}_Y(Y \notin s)) \left( \sum_{y \in s} v_y - \sum_{y \notin s} v_y \right).$$

Hence, without any context variable, the min-max game (8) can be rewritten as

$$\min_{v \in \Delta_{\mathcal{Y}}} \max_{\mu \in \Delta_{\mathcal{S}}} - \sum_{s \in \mathcal{S}} \mu_s (\mathbb{P}_Y(Y \in s) - \mathbb{P}_Y(Y \notin s)) \left( \sum_{y \in s} v_y - \sum_{y \notin s} v_y \right). \qquad (30)$$

We will analyze this problem through the lens of a mix-actions zero-sum game. We know from von Neumann and Morgenstern (1944) that a solution to this min-max problem exists, and that one can switch the min-max to a max-min without modifying the value of the solution. Let us denote by $(v^*, \mu^*)$ the argument of a solution. To minimize the value of this game, the player $v$ should play such that

$$\operatorname{sign}(\sum_{y \in s} v_y^* - \sum_{y \notin s} v_y^*) = \operatorname{sign}(\mathbb{P}(Y \in s) - \mathbb{P}(Y \notin s)) = \operatorname{sign}(\sum_{y \in s} \mathbb{P}(Y = y) - \sum_{y \notin s} \mathbb{P}(Y = y)),$$

which allows this player to ensure a negative value to the game. Stated otherwise

$$\forall s \in \mathcal{S}, \qquad \mathbb{P}(Y \in s) > \frac{1}{2} \quad \Rightarrow \quad \sum_{y \in s} v_y^* \geq \frac{1}{2}. \qquad (31)$$

As a consequence, if there exists any set such that $\mathbb{P}(Y \in s) = 1/2$, the best strategy of player $\mu$ is to play only those sets to ensure the value zero, and any $v$ that satisfies (31) is optimal. It should be noted that (31) does not generally imply that $(v_y)_{y \in \mathcal{Y}}$ has the same ordering as $(\mathbb{P}(Y = y))_{y \in \mathcal{Y}}$.

When $\{y^*\} \in \mathcal{S}$ and $\mathbb{P}(Y = y^*) > 1/2$, if $v = \delta_{y^*}$, the prediction player is able to ensure a value of $\max_{s \in \mathcal{S}} - |2 \mathbb{P}(Y \in s) - 1|$, which is maximized by the query player with $s = \{y^*\} \cup s'$ for any $s'$ such that $\mathbb{P}(Y \in s') = 0$. Other strategies for $v$ will only increase this value, hence $v^* = \delta_{y^*}$ which implies the first part of Proposition 4.

**A counter example.** While we hope that the solution $(v^*, \mu^*)$ does characterize the original solution $y^*$, it should be noted that $v^*$ alone does not characterize $y^*$. Indeed, it is even possible to have $v^*$ uniquely defined without having $y^* = \arg\max_{y \in \mathcal{Y}} v_y^*$. For example, consider the case where $\mathcal{Y} = \{1, 2, 3\}$ and $(\mathbb{P}(Y = i))_{i \in [3]} = (.4, .3, .3)$. By symmetry, the player $\mu$ only has to play on $\mathcal{S} = \{\{1\}, \{2\}, \{3\}\}$, which leads to the min-max game

$$\min_v \max_\mu \begin{pmatrix} \mu_{\{1\}} \\ \mu_{\{2\}} \\ \mu_{\{3\}} \end{pmatrix}^\top \begin{pmatrix} .2 & -.2 & -.2 \\ -.4 & .4 & -.4 \\ -.4 & -.4 & .4 \end{pmatrix} \begin{pmatrix} v_1 \\ v_2 \\ v_3 \end{pmatrix}.$$

The value of this game is $-.1$ and is achieved for $\mu^* = (.5, .25, .25)$, $v^* = (.25, .375, .375)$.

### D.2 Optimization procedure

Let us rewrite the problem through the objective

$$\mathcal{E}(g, \mu) = \mathbb{E}_{(X,y) \sim \rho} \mathbb{E}_{S \sim \mu(x)} [L(g(X), S, \mathbf{1}_{Y \in S} - \mathbf{1}_{Y \notin S})].$$

We want to solve the min-max problem $\min_g \max_\mu \mathcal{E}(g, \mu)$. This problem can be solved efficiently based on the vector field point of view of gradient descent (Bubeck, 2015) if:

- we can parametrize the function $g : \mathcal{X} \to \Delta_{\mathcal{Y}}$ such that $\mathcal{E}$ is convex with respect to the parametrization of $g$;
- we can access unbiased stochastic gradients of $\mathcal{E}$ with respect to $g$ that have a small second moment;
- we can parametrize the function $\mu : \mathcal{X} \to \Delta_{\mathcal{S}}$ such that $\mathcal{E}$ is concave with respect to the parametrization of $\mu$;
- we can access unbiased stochastic gradients of $\mathcal{E}$ with respect to $\mu$ that have a small second moment.

The first two points are no problems, $g$ can be parametrized with softmax regression, and since $L$ is linear with respect to the scores, it will keep the problem convex. Moreover, to access a stochastic gradient of $\mathcal{E}$, one can sample $X_i \sim \rho_{\mathcal{X}}$ and $S_i \sim \mu(X_i)$ before querying $\mathbf{1}_{Y_i \in S_i}$ and computing the gradient of $L(g(X_i), S_i, \mathbf{1}_{Y_i \in S_i} - \mathbf{1}_{Y_i \notin S_i})$ with respect to the parametrization of $g$.

The third point is slightly harder to tackle. Since $\mathcal{E}$ is linear with respect to $\mu$, one way to proceed is to find a linear parametrization of $\mu$. In particular, one can take a family $(g_i)_{i \in [N]}$ of linearly independent

functions from $\mathcal{X}$ to $\Delta_{\mathcal{S}}$ and search for $g$ under the form $\sum_{i \in [N]} c_i g_i$ for $(c_i)$ positive summing to one. To build such a family, one can eventually use "atom functions" and simple operations such as symmetry with respect to $\mathcal{Y}$ and $\mathcal{S}$, rescaling, translation, rotations with respect to $\mathcal{X}$. For example if $\mathcal{X}$ is a Banach space, one could define atom functions as, for $y_i \in \mathcal{Y}$

$$g_i : x \rightarrow \frac{\|x\|}{1 + \|x\|} \frac{1}{|\mathcal{S}|} \sum_{s \in \mathcal{S}} e_s + \frac{1}{1 + \|x\|} e_{\{y_i\}}.$$

Those functions could be rescaled and translated as $g_{\sigma,\tau,i}(x) = g_i(\sigma(x - \tau))$, in order to specify a family $(g_{\sigma,\tau,i})$ from few values for $\tau$ and $\sigma$.

The last point is the most difficult one. Without context variables, and with no-parametrization for $\mu$, a naive unbiased gradient strategy for $\mu$ consists in asking random questions to update the full knowledge of $(\mathbb{P}(Y \in s))_{s \in \mathcal{S}}$. But such a strategy will be much worse than our median surrogate technique with queries $\mathbf{1}_{Y \in \{y\}}$ for $y$ sampled uniformly at random in $\mathcal{Y}$. Eventually, one should go for a biased gradient strategy, while making sure to update $\mu$ coherently to avoid getting stalled on bad estimates as a result of biases.

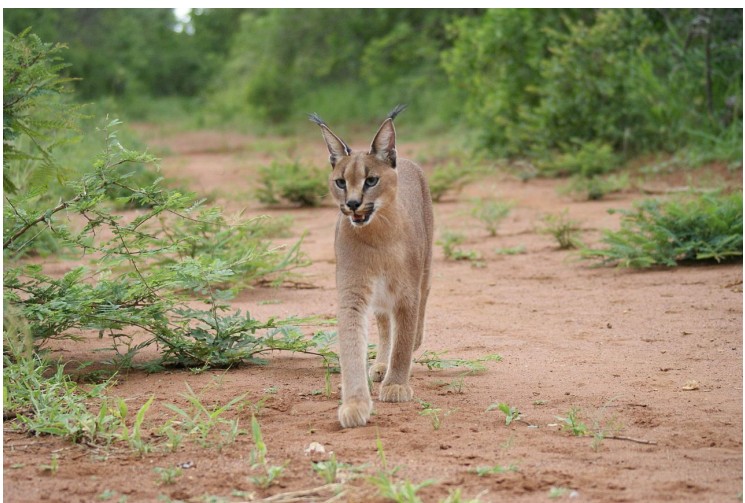

**Figure 6:** Recognizing fine-grained classes is difficult, but recognizing attributes is easy.

# E    Experimental details

Our experiments are done in *Python*. We leverage the *C* implementation of high-level array instructions by Harris et al. (2020), as well as the visualization library of Hunter (2007). Randomness in experiments is controlled by choosing explicitly the seed of a pseudo-random number generator.

## E.1    Comparison with fully supervised SGD

In this section, we investigate the difference between weakly and fully supervised SGD. According to Theorem 1, we only lost a constant factor of order $m^{3/2}$ in our rates compared to fully supervised (or plain) SGD. This behavior can be checked by adding the plain SGD curve on Figure 2. On the left side of Figure 8, we do observe that the risk of both Algorithm 1 and plain SGD decrease with same exponent with respect to number of iteration but with a different constant in front of the rates: that is we observe the same slopes on the logarithm scaled plot, but different intercepts. Going one step further to check the tightness of our bound, one can plot the intercept, or the error achieved by both Algorithm 1 and plain SGD as a function of the output space dimension $m$. The right side of Figure 8 shows evidence that this error grows as $m^\varepsilon$ for some $\varepsilon \in [1, 3/2]$, which is coherent with our upper bound. Similarly to Figure 2, this figure was computed after cross validation to find the best scaling of the step sizes for each dimension $m$.

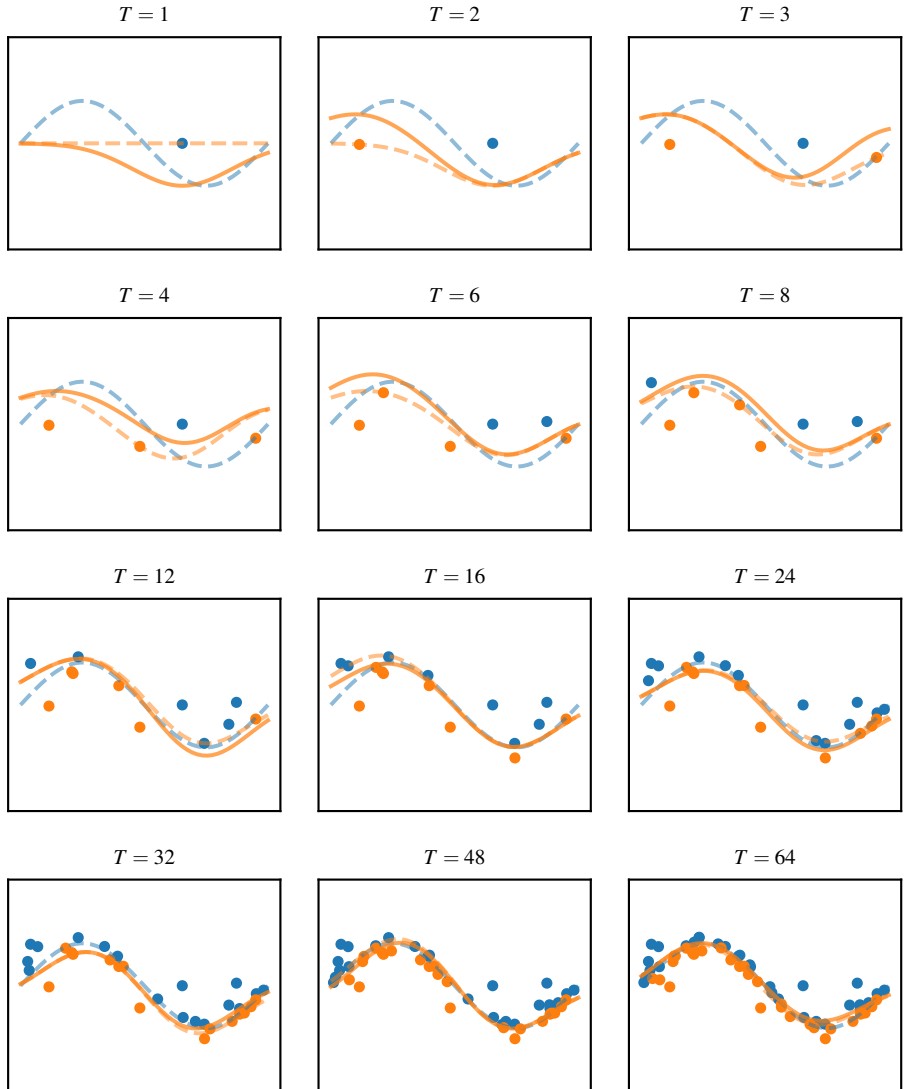

**Figure 7:** *Streaming history of the active strategy* to reconstruct the signal in dashed blue in the same setting as Figure 1. At any time $t$, a point $X_t$ is given to us, our current estimate of $\theta_t$ plotted in dashed orange gives us $z = f_{\theta_t}(X_t)$, and we query $\text{sign}(Y_t - z)$. Based on the answer to this query, we update $\theta_t$ to $\theta_{t+1}$ leading to the new estimate of the signal in solid orange. In this figure, we see that it might be useful for the practitioners in a streaming setting to reduce the bandwidth of $\varphi$ as they advance in time.

## E.2 Passive strategies for classification

A simple passive strategy for classification based on median surrogate consists in using the active strategy with coordinates sampling, that is $u$ being uniform on $\{e_y\}_{y \in \mathcal{Y}}$, where $(e_y)_{y \in \mathcal{Y}}$ is the canonical basis of $\mathbb{R}^{\mathcal{Y}}$ used to define the simplex $\Delta_{\mathcal{Y}}$ as the convex hull of this basis. Querying $\mathbf{1}_{\langle g_\theta(x) - e_y, e_y \rangle > 0}$ is formally equivalent to the query of $\mathbf{1}_{Y=y}$ when $g_\theta(x) \in \Delta_{\mathcal{Y}}$. This is the baseline we plot on Figure 2.

A more advanced passive baseline is provided by the infimum loss (Cour et al., 2011; Cabannes et al., 2020). It consists in solving

$$\arg\min_{f:\mathcal{X}\to\mathcal{Y}} \mathcal{R}_I(f) := \mathbb{E}_{(X,Y)\sim\rho} \, \mathbb{E}_S \left[ L(f(X), S, \mathbf{1}_{Y\in S}) \right],$$

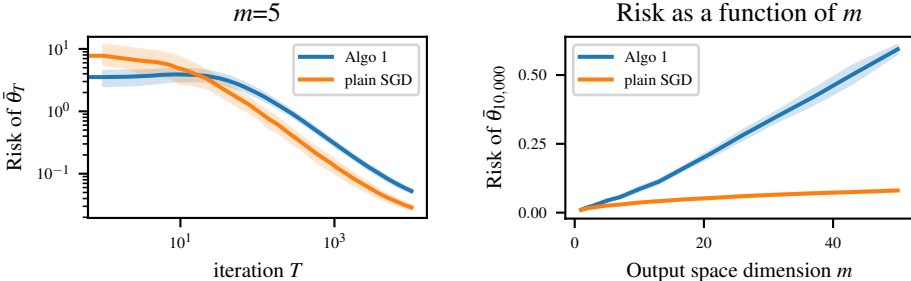

**Figure 8:** *Comparison of generalization errors of weakly and fully supervised SGD* as a function of the annotation budget $T$ and output space dimension $m$. The setting is similar to Figure 2. We observe a transitory regime before convergence rates follows the behavior described by Theorem 1. The right side plots the error of both procedures after 10,000 iterations as a function of the output space dimension $m$ between 1 and 50. The number of iteration ensures that, for all values of $m \in [50]$, the reported error is well characterized by our theory, in other terms that we have entered the regime described by Theorem 1.

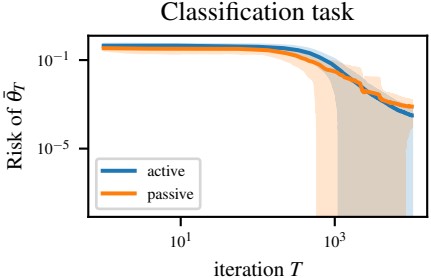

**Figure 9:** *Comparison with the infimum loss with better conditioned passive supervision* in a similar setting to Figure 2 yet with $m = 10$, $\varepsilon = 0$, that is $X$ uniform on $\mathcal{X}$, and $\gamma_0 = 7.5$ for the active strategy and $\gamma_0 = 15$ for the passive strategy. We see no major differences between the active strategy based on the median surrogate and the passive strategy based on the median surrogate with the infimum loss. Note that the standard deviation is sometimes bigger than the average of the excess of risk, explaining the dive of the dark area on this logarithmic-scaled plot.

where $S$ is a random subset of $\mathcal{Y}$ and $L$ is defined from the original loss $\ell : \mathcal{Y} \times \mathcal{Y} \to \mathbb{R}$ as, for $z \in \mathcal{Y}$, $s \subset \mathcal{Y}$ and $y \in \mathcal{Y}$,

$$L(z, s, \mathbf{1}_{y \in s}) = \begin{cases} \inf_{y' \in s} \ell(z, y') & \text{if } y \in s \\ \inf_{y' \notin s} \ell(z, y') & \text{otherwise.} \end{cases}$$

Random subsets $S$ could be generated by making sure that the variable $(y \in S)_{y \in \mathcal{Y}}$ are independent balanced Bernoulli variables; and by removing the trivial sets $S = \emptyset$ and $S = \mathcal{Y}$ from the subsequent distribution. In order to optimize this risk in practice, one can use a parametric model and a surrogate differentiable loss together with stochastic gradient descent on the empirical risk. For classification with the 0-1 loss, we can reuse the surrogate introduced in Proposition 3 and minimize, assuming that we always observed $\mathbf{1}_{Y_i \in S_i} = 1$ for simplicity,

$$\hat{\mathcal{R}}_{I,S}(\theta) = \sum_{i=1}^{n} \inf_{y \in S_i} \left\| g_\theta(X_i) - e_y \right\|.$$

Stochastic gradients are then given by, assuming ties have no probability to happen,

$$\nabla_\theta \inf_{y \in S_t} \left\| g_\theta(X_t) - e_y \right\| = \left( \frac{g_\theta(X_t) - e_{y^*}}{\left\| g_\theta(X_t) - e_{y^*} \right\|} \right)^\top Dg_\theta(X_t) \quad \text{with} \quad y^* := \underset{y \in S_t}{\arg\max} \left\langle g_\theta(X_t), e_y \right\rangle.$$

This gives a good passive baseline to compare our active strategy with. In our experiments with the Gaussian kernel, see Figure 9 for an example, we witness that this baseline is highly competitive.

Although we find that it is slightly harder to properly tune the step size for SGD, and that the need to compute an argmax for each gradient slows-down the computations.

## E.3 Real-world classification datasets

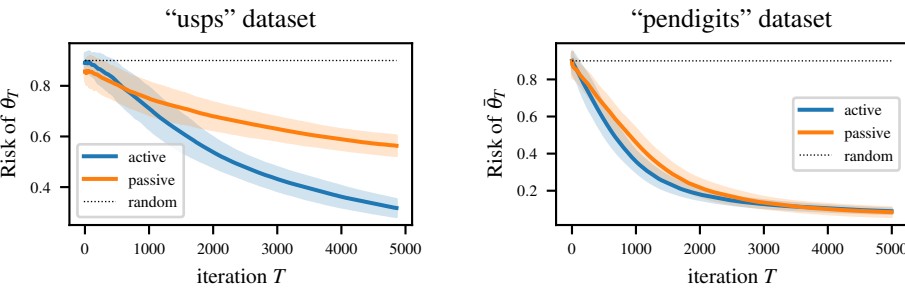

**Figure 10:** *Testing errors on two LIBSVM datasets* with a similar setting to Figure 9. Those empirical errors are reported after averaging over 100 different splits of the datasets. The step size parameter was optimized visually, which led to $\gamma_0 = 15$ for the active strategy on "USPS", $\gamma_0 = 60$ for the passive one, $\gamma_0 = 7.5$ for the active strategy on "pen digits", $\gamma_0 = 30$ for the passive one. The dotted line represents $\mathcal{R} = 1 - m^{-1}$ which is the performance of a random model.

In Figure 10, we compare the "well-conditioned" passive baseline with our active strategy on the real-world problems of LIBSVM (Chang and Lin, 2011). We choose the "USPS" and "pen digits" datasets as they contain $m = 10$ classes each with $n = 7291$ and $n = 7494$ samples respectively, with $d = 50$ and $d = 16$ features each. We have chosen those datasets as they present enough classes that leads to many different sets $S$ to query, and they are made of the right number of samples to do some experiments on a laptop without the need for "advanced" computational techniques such as caching or low-rank approximation (Meanti et al., 2020). On Figure 10, we use the same linear model as for Figure 2, that is a Gaussian kernel. We choose the bandwidth to be $\sigma = d/5$, and we normalize the features beforehand to make sure that they are all centered with unit variance. We report error by taking two thirds of the samples for training and one third for testing, and averaging over one hundred different ways of splitting the datasets. We observe that the active strategy leads to important gains on the "USPS" dataset, yet is not that useful for the "pen digits" dataset. We have not dug in to understand those two different behaviors.

## E.4 Real-world regression dataset & Nyström method

In this section, we provide two experiments on real-world datasets.

In order to deal with big regression datasets, it is useful to approximate the parameter space $\mathcal{Y} \otimes \mathcal{H}$ in Assumption 1 with a small dimensional space. To do so, let us remark that given samples $(X_i)_{i \leq n} \in \mathcal{X}^n$ for $n \in \mathbb{N}$, we know that our estimate $f_{\theta_n}$ can be represented as

$$f_{\theta_n}(\cdot) = \sum_{i \leq n} \sum_{j \leq m} a_{ij} \langle \varphi(x_i), \varphi(\cdot) \rangle e_j,$$

for some $(a_{ij}) \in \mathbb{R}^{p \times m}$ and where $(e_j)_{j \leq m}$ is the canonical basis of $\mathcal{Y} = \mathbb{R}^m$. For large datasets, that is when $n$ is large, it is smart to approximate this representation through the parameterization

$$f_a(x) = \sum_{i \leq p} \sum_{j \leq m} a_{ij} k(x, x_i) e_j,$$

where $p \leq n$ is the rank of our approximation, and $k$ is the kernel defined as $k(x, x') = \langle \varphi(x), \varphi(x') \rangle$. Stated with words, we only use a small number $p$, instead of $n$, of vectors $\varphi(x_i)$ to parameterize $f$. This allows to only keep a matrix of size $p \times m$ in memory instead of $n \times m$, while not fundamentally changing the statistical guarantee of the method (Rudi et al., 2015). In this setting, the stochastic gradients are specified from the fact that

$$u^\top D_a f_a(x) = (u_j k(x, x_i))_{i,j} \in \mathbb{R}^{p \times m}.$$

In other terms, in order to update the parameter $a$ with respect to the observation made at $(x, u)$, we check how much each coordinate of $a$ determines the value of $u^\top f_a(x)$.

In the following, we experiment with two real-world datasets. In order to learn the relation between inputs and outputs, we use a Gaussian kernel after normalizing input features so that each of them has zero mean and unit variance. To keep computational cost, we sample $p$ random (Nyström) representers among the training inputs which are used to parameterize functions. To avoid overfitting, we add a small regularization to the empirical objective. It reads $\lambda \|\theta\|_{\mathcal{H}}^2$ with our notations and corresponds to the Hilbertian norm inherited from the reproducing kernel $k$ of the function $f_\theta$ (Scholkopf and Smola, 2001).

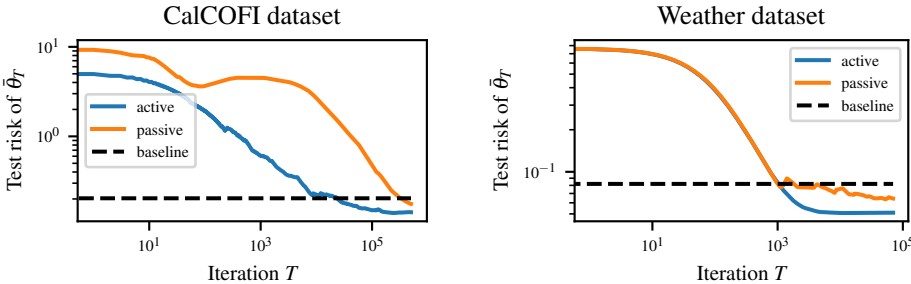

**Figure 11:** *Testing error on two real-world regression datasets.* On both datasets, a single pass was made through the data in a chronological fashion, and errors were computed from the 26,453 most recent data samples for the "Weather" dataset, and from a random sample of 10,000 samples among the 155,140 most recent samples for the "CalCOFI" dataset.

Our first experiment is based on the data collected by the California Cooperative Oceanic Fisheries Investigation between March 1949 and November 2016.[6] It consists of more than 800,000 seawater samples including measurements of nutriments (set aside in our experiments) together with pressure, temperature, salinity, water density, dynamic height (providing five input parameters), as well as dissolved oxygen, and oxygen saturation (the two outputs we would like to predict). We assume that we can measure if any weighted sum of oxygen concentration and saturation is above a threshold by letting some population of bacteria evolves in the water sample and checking if it survives after a day. If the measurements are done on the day of the sample collection, this setting exactly fits in the streaming active labeling framework. After cleaning the dataset for missing values, the dataset contains 655,140 samples. The "CalCOFI" dataset results are reported on the left of Figure 11, parameters were chosen as $p = 100$, $\sigma = 10$, $\lambda = 10^{-6}$ and $\gamma_0 = 1$. For the passive strategy, random queries were chosen to follow a normal distribution with the same mean as the targets and one third of their standard deviation (*i.e.* we ask if the apparent temperature is lower than the usual one plus or minus a perturbation). The plotted baseline corresponds to linear regression performed over the entire dataset. It takes about 10,000 samples for our active strategy to be competitive with this baseline, and 200,000 samples for the passive one.

The second experiment makes use of data collected through the Dark Sky API (which is now part of Apple WeatherKit). It is made of 96,454 weather summaries between 2006 and 2016 in the city of Szeged, Hungary. Our task consists in computing the apparent temperature from real temperature, humidity, wind speed, wind bearing, visibility and pressure. The apparent temperature is an index that searches to quantify the subjective feeling of heat that humans perceive, it is expressed on the same scale as real temperature. One way to measure it would be to ask some humans if the outside is hotter or colder than a controlled room with a specific temperature and neutral meteorological conditions. Once again, this exactly fits into our streaming active labeling setting. The "Weather" dataset results are reported on the right of Figure 11. The baseline consists in predicting the apparent temperature as the real temperature. We observe a transitory regime where the first 1,000 samples seem to be used to calibrate the weights $\alpha$. During this regime, our estimate is too bad for the active strategy to make smarter queries than the "random" ones that have been calibrated on temperature statistics. The main difference in the learning dynamic between the active and passive strategies is

---

[6]CalCOFI data is licensed under the CC BY 4.0 license and the data is available at `https://calcofi.org/`.

observed on the remaining 69,000 training samples. The parameters were the same as the "CalCOFI" dataset but for $\gamma_0 = 10^{-2}$.