# OpenReview forum: "Active Labeling: Streaming Stochastic Gradients"
_NeurIPS.cc/2022/Conference — NeurIPS 2022 Accept_

### Official Review · Reviewer_yumH · 2022-07-09

**Rating:** 6
**Confidence:** 3
**Soundness:** 4 excellent
**Presentation:** 3 good
**Contribution:** 2 fair

**Summary:**

The authors (re-)coins the term "active labeling" as a general case of Active-Learning where to the case of weak/partial supervision.
During training, the proposed method requires access only to the (stochastic) gradients deduced from the annotation, which allows some degree of privacy preserving training if used in a streaming regime.
Since the problem statement might seem too abstract, the authors provide three concrete real-life examples for partial annotation in lines 78-98.
The authors choose, however, to exemplify the approach for the specific case of (robust) median regression where the supervision partiality is half-planes.

**Questions:**

The median regression problem is indeed a worthy one.
Just to gain some sense on the next steps - what is the gap to present results on one of the three examples (or any other similar real-life problem)?

**Limitations:**

Even though the authors claim they state the limitations of their method in the discussion section, I did not find any that apply to the specific method in the text, but rather to the lager problem of active learning or continuous gradient based optimization.

**Strengths And Weaknesses:**

The paper is rigorous and very well written, even if some parts are too math-intensive than needed (e.g. lines 121-123 might be easier to understand with some context or more verbosity).
The theoretic section seems sound - I read through some of the proofs and did not find any corrections or mistakes.

This is a good paper, imho, so any weaknesses I will point our might seem superficial:
As mentioned, the examples in lines 78-98 really give motivation. However, the particular case that the authors chose to address (median regression) is a bit underwhelming in comparison, and a bit harder to apply to real-life. Even more so, I think any one of the three examples are easy to test by holding out data from open datasets (like imagenet).

---

> ### Author Response · Authors · 2022-07-31
> **Answer to Reviewer yumH**
>
> Thank you very much for your appreciation of our work, as well as for the time you took to review and understand our work.
>
> We are sorry if some passages were a bit too dry: stated with words, lines 121-123 states that if you have a way to image a vector $\theta$ from partial measurement $1_{\theta\in T}$ such that you can reconstruct this vector in a linear fashion, this is equation (3), then it provides you a generic strategy to get an unbiased stochastic estimate of this vector from a partial measurement, this is equation (2).
> To do so, we voluntarily introduce randomness over our measurements.
> We have integrated this comment into our manuscript.
> Thank you for spotting this passage and helping us improve the readability of our work.
> Please do not hesitate to point us to other parts of the manuscript that could be hard to parse for the reader.
>
> The main reason why we did not want to extend too much on discrete output problems is because we hope to design future algorithms that would better leverage the discrete output structure more smartly (retaking ideas from combinatorial bandit for classification and from active ranking in the context of preference learning).
> However, nothing prevents us from using our algorithm on those problems.
> We understand that this paper would have a greater impact had we shown that this is not only a proof-of-concept but that it does work on real-world data.
> As a consequence, we provided real world experiments in the rebuttal file in order to convince you that this work is not only theory that sounds good, but also a practical algorithm that can be helpful in the real world.
> You will find those in Appendix E.
>
> As a conclusion, we would like to thank you for your appreciation of this work, and we stay available for further discussion or precision.

---

> > ### Comment · Reviewer_yumH · 2022-08-08
> > **Thank you for your response**
> >
> > .

---

### Official Review · Reviewer_Byvs · 2022-07-12

**Rating:** 6
**Confidence:** 4
**Soundness:** 3 good
**Presentation:** 3 good
**Contribution:** 3 good

**Summary:**

This paper introduces the ``active labeling'' problem, which aims at learning with weakly-supervised labeling information. An specific focus of the paper is to conduct SGD but without the full gradient. The authors show that how to query weakly-supervised information in the cases of least-squares and the median regression. The authors also provide minimax results regarding the convergence rate (in the online setting). Empirical evaluations show the efficacy of the proposed method compared to the passive counterpart.

**Questions:**

Since theorem 2 essentially says that the provided Algorithm 1 works as good as the SGD with full gradient information, I wonder if the authors can conduct an experiment to compare the performance of Algorithm 1 and standard SGD with full gradient information?

**Limitations:**

The authors discussed the limitation in Section 8.

**Strengths And Weaknesses:**

Strengths:
1. A newly proposed ``active labeling'' problem looks interesting and important.
2. The authors provide an algorithm to handle the proposed problem under median regression, and prove its minimax optimality in the kernel/linear case.

Weakness:
1. While I found the discussion in Section 7 quite interesting, I believe it will be better if precise statements are provided as theorems/propositions, e.g., on line 267, the authors claim exponential convergence rates can be achieved under margin conditions but no formal theorems are provided.

---

> ### Author Response · Authors · 2022-07-31
> **Answer to Reviewer Byvs**
>
> We are deeply thankful for the valuable time you took to review our work, we are happy to read that you have appreciated it.
>
> We have happily conducted an experiment to illustrate the difference of Algorithm 1 with SGD. It is in the rebuttal file (in Appendix E.1).
> In theory, this difference can be read in the constant in front of the convergence rates in Theorem 1 (in particular our constant reads $\kappa M m^{3/2}$ when it reads $\kappa M$ for full SGD).
>
> We are glad you asked for precision about the exponential convergence rates.
> Such rates of convergence can be proven under the following low-noise assumption (known as Massart noise condition, or Tsybakov hard margin): there exists a threshold $\delta > 0$ such that for almost all $x\in{\cal X}$ and $z\in {\cal Y}$, and $\mathbb{P}(Y = f(x) \vert X=x) - \mathbb{P}(Y = z\vert X=x) \notin (0, \delta)$.
> For example, this assumption is met when for each input $X$ the most probable class has always more than 60\% of chance to be the target $Y$. Arguably, this is true for well-curated images dataset such as ImageNet or CIFAR10.
> Together with Assumption 1, that is if the surrogate target $g^*$ belongs to the RKHS and the kernel is bounded, then the right hand-side of equation (6) can be replaced by $\exp(-cT)$ for some constant $c$.
> The proof would be a simple adaptation of the work of Pillaud-Vivien *et al,* *Exponential Convergence of Testing Error for Stochastic Gradient Methods* (2018) to our case.
>
> We stay at your disposal for any further concerns or questions.

---

> > ### Comment · Reviewer_Byvs · 2022-08-06
> > **Thanks for your response**
> >
> > Thank you for your response. The exponential speedup under Massart noise certainly makes sense; however, I still think it's better to formally add a proposition/corollary in the paper regarding the exponential speedups.
> >
> > Also, I wonder if it's better to formally introduce the problem as ''active weakly supervised learning'' rather than the ambiguous term ''active labeling'' (as discussed in footnote 1; I also see other reviewers using wordings like ''re-coin the term 'active labeling''', which can lead to confusions).

---

> > > ### Comment · Reviewer_yumH · 2022-08-08
> > > **the term active labeling**
> > >
> > > I agree that ''active weakly supervised learning'' delivers the message better (even if a bit wordy)

---

> > > > ### Author Response · Authors · 2022-08-09
> > > > **Authors' answer**
> > > >
> > > > You are right, we will happily add a formal proposition about exponential convergence rates, and it is true that the renaming "active labeling" into "active weakly supervised learning" might help people interested in active weakly supervised learning to more easily find our work. Thank you for this suggestion.

---

### Official Review · Reviewer_Evw4 · 2022-07-12

**Rating:** 5
**Confidence:** 4
**Soundness:** 3 good
**Presentation:** 3 good
**Contribution:** 3 good

**Summary:**

This paper has two main contributions. First, the paper defines a problem setting known as "active labeling", that is, given a large output space $\mathcal{Y}$, we can query an index $i \in [n]$ of a dataset with a subset $S \subset \mathcal{Y}$ and receive the binary response of whether $y_i \in S$ or $y_i \not\in S$. The paper gives a few compelling examples that fit in this setting and a generic way to produce sets $S$ to enable SGD. Second, the paper investigates two specific settings, least-squares regression and median regression, and provides an algorithm with analysis and a matching (in terms of dependence on number of samples) lower bound.

**Questions:**

I understand the "active labeling" setting is general and well-motivated. Can the authors provide a realistic use case of the algorithmic contributions?

I wonder if this paper would be better written as a privacy paper without mention of "acquiring the most informative dataset". In the privacy preserving setting, the "streaming setting" is very realistic (as the authors point out) and the arbitrary sets $S$ is realistic as a user's device can evaluate if $y_i \in S$ (no need for human computation). As written, the paper's area and scope seem a bit scattered.

**Limitations:**

Yes

**Strengths And Weaknesses:**

Strengths:
 - This paper provides a nice, meaningful analysis of median regression in the "active labeling" setting.
 - The paper is overall clear and easy to understand (perhaps with the exception of Section 3 which I feel could be expanded).

Weaknesses:
 - While the motivation for "active labeling" is compelling, the algorithmic contributions seem to require arbitrary sets $S$ while the compelling examples hinge on "a specified set of subsets of $\mathcal{Y}$ ". For example, the classification with attributes example does not allow all 2^{number of attributes} subsets to be queried. It appears that the construction of stochastic gradients in Section 3 almost requires arbitrary queries.
 - It seems to me that a concrete use case of the algorithm is not fleshed out. In most papers, the experiments section ensures this, but the experiments are quite synthetic (there is "real-world" data in appendix E but perhaps not a realistic query model).
 - I'm not sure the setting "generalizes active learning based on partial supervision" and furthermore, the "streaming technique" is in a rather different setting (requiring information from each point in the stream) compared to standard streaming active learning.

---

> ### Author Response · Authors · 2022-07-31
> **Answer to Reviewer Evw4**
>
> We would like to warmly thank you for your constructive comments and your detailed concerns that will help us improve our draft.
> We hope that we will succeed in convincing you of the quality and usefulness of our work.
>
> We are happy to read that we succeed in presenting our message in a clear and easy-to-understand manner.
> Following your comment as well as the one of reviewer yumH, we have rephrased the technical equations of Section 3 with words in the rebuttal file.
> You are right that we do not generalize active learning *stricto sensu*, we rather provide a variant based on "weak supervision". We have modified our wording in the abstract.
>
> It is true that our SGD procedure could have strong applications for privacy preserving issues, yet our main motivation was to define the "active labeling" problem (which has been on our mind for a while).
> Hence, we would be keen to keep the paper as it is, that is a presentation of this "active labeling" setup, that we found abstract enough for theory to be generic, but concrete enough to have a clear impact (plus a consistent algorithm to tackle it).
> We indeed believe that the definition of the "active labeling" problem is already a fair contribution.
>
> Among the different approaches we have investigated, the SGD solution has the advantages of being robust to noise, easy to implement, quite generic and to satisfy some minimax optimality properties, but you are right that it does not easily deal with restriction on the sets to query (although Proposition 4 provides an option for classification with attributes). We have made this point clearer in the rebuttal file.
> To extend on the "arbitrary queries", let us stress out that our SGD procedure leads to queries that follow the output geometry induced by the loss. To take a concrete example, ranking problems can be approached with correlation losses (Kendall's tau, Spearman's rho, Hamming loss) and tackled through surrogate regression problems where the output spaces are the convex hulls of some well-known polytopes (see the work of Nir Ailon or Anna Korba for example), such as the Birkhoff polytope or the permutohedron.
> Although their descriptions is out-of-scope out this paper, linear cut of those polytopes are not completely arbitrary set.
> For example, the faces of all dimensions of the permutohedron correspond, in a one-to-one fashion, to strict weak orderings (*e.g.* Thompson (1993) *Generalized Permutation Polytopes and Exploratory Graphical Methods for Ranked Data*).
>
> We understand that, since we mainly focus on the regression case, and in order to provide realistic use cases of our algorithmic contributions, we could have motivated it with more real-world examples beside the pricing example.
> To do so, let us give a practical example from real life : our setting could be useful in a situation where one acquire tissues through an invasive biopsy in order to quantify the concentration of some specific elements (such that the amount of connective tissue in parenchymal tissue to check for fibrosis scarring).
> Suppose that they can proceed for this measurement by cutting the tissue in a few pieces and putting them into different levels of reactive solvent that would turn to a specific color if a threshold is met.
> This exactly fits into our regression framework with the observation of half spaces.
> You can generalize this example to any situations where you have sensors that only give a binary answer with respect to the measure of a continuous quantity.
> There is many other examples where one has to measure continuous values and can only acquire partial information, this has been a long-standing problem in economy following the seminal work of James Tobin (one can also excavate works in physics that date back to the 19th century such as the paper of William Sheppard *On the calculation of the most probable values of frequency constants, for data arranged according to equidistant division of a scale.*).
>
> We also hear that the experimental part seems too much like a proof-of-concept and not enough like real experimental validation.
> Therefore, we provide empirical validation on two real-world datasets found on Kaggle (without curating dataset, only looking for the first results on the Web).
> The results have been added in the Appendix E.4 of the rebuttal revision.
>
> We would like to thank you again for having committed to review our work, and we stay at your disposal to discuss or answer any question you might have.

---

> > ### Comment · Reviewer_Evw4 · 2022-08-03
> > **Reviewer's response**
> >
> > Thank you for the example of ranking problems. Although I didn't work it out, I'd be surprised if all "linear cuts of those polytopes" are interpretable to humans (such as an ordering of a few items as mentioned as Example 2 in the paper).
> >
> > Thank you for the concrete examples of tissue samples, oxygen levels in water, and apparent temperature. After thinking about it a bit, I think if the output space is one-dimensional, then thresholding by a value is a reasonable practical setting. However, I think that observing a halfspace for a multi-dimensional output space is not reasonable. I cannot easily think of any cases where one can measure if a weighted combination of outputs meets a threshold.
> >
> > For the tissue samples, it is written: "Suppose that they can proceed for this measurement by cutting the tissue in a few pieces and putting them into different levels of reactive solvent that would turn to a specific color if a threshold is met. This exactly fits into our regression framework with the observation of half spaces." I'm not sure I quite understand the setup, but my best interpretation either yields observing thresholds on each output (axis-aligned half spaces) or some sort of weighted sum chemistry that seems implausible.
> >
> > For the oxygen levels in water, it is written: "We assume that we can measure if any weighted sum of oxygen concentration and saturation is above a threshold by letting some population of bacteria evolves in the water sample and checking if it survives after a day." Is there a species of bacteria that has a clean linear classifier for death as a function of oxygen concentration and oxygen saturation? Furthermore, is there a species of bacteria for every linear classifier (or at least an epsilon-net or something)? I find this somewhat implausible.
> >
> > I liked the apparent temperature setting is a good use case, but perhaps only because it involves a one-dimensional output space.
> >
> > -------------------------------------
> >
> > I would like to point out that the optimality of the algorithm is only in terms of the dependence on T. It looks like the dependence on the dimension is worse, which I think is expected because of the less informative types of queries.
> >
> > -------------------------------------
> >
> > It seems to me that perhaps the "active labeling" general problem lacks enough structure to create effective algorithms, and thus might not be the right abstraction. For example, the algorithms and methods presented require rather general query sets while the practical use cases have strong restrictions on the query sets. I concede that the setting of linear models and one-dimensional outputs (observing thresholds) is reasonable, but this is quite narrow.
> >
> > --------------------------------------
> >
> > I still find the story/scope of the paper scattered without a clear, useful message.

---

> > > ### Author Response · Authors · 2022-08-03
> > > **Discussion with Reviewer Evw4**
> > >
> > > Thank you for your responsiveness, let us try one more time to convince you of the usefulness of our work.
> > >
> > > >  I'd be surprised if all "linear cuts of those polytopes" are interpretable to humans.
> > >
> > > Since a linear cut of a polytope is a union of faces of different dimensions, linear cuts of the permutohedron are made of unions of weak orderings.
> > > Note also that our imaging strategy Eq. (5) can be adapted to cases where $U$ is not uniform on the sphere but satisfies some isotropic properties.
> > > Although out of scope of this paper, we believe that one can leverage isotropy to reduce queries to small unions of weak orderings that are easy to interpret for humans.
> > > We indeed give some concrete details in our answer to Reviewer 1mW2 on one way to approach ranking with our algorithm in order to learn based on pairwise orderings.
> > >
> > > >  I cannot easily think of any cases where one can measure if a weighted combination of outputs meets a threshold.
> > >
> > > We understand that the example of oxygen concentration plus saturation we built from the CalCOFI dataset might sound a bit superficial: of course, no one will measure oxygen concentration level with bacteria since there are excellent cheap ways to measure oxygen with titration methods.
> > > Sadly, we are not knowledgeable enough in chemistry and biology to know of an analogical sensor that can measure any weighted sum of two (or more) quantities like on our oxygen concentration plus saturation example.
> > > Yet, compressed sensing photography gives a "reasonable" example where one can measure if any weighted combination of outputs meets a threshold (*e.g.* Dadkhah *et al.* (Sensors, 2013) *Compressive Sensing Image Sensors-Hardware Implementation*).
> > > This setting consists in acquiring an image made of $m$ pixels with a unique optical sensor.
> > > Usually a camera is made of $m$ optical sensors that measure the light intensity of each pixel.
> > > Instead, one can put small reflectors where the pixel sensors were, that all lead to a unique optical sensor.
> > > The sensor will then only collect information about the summation of light intensity of pixels whose reflectors were on.
> > > By switching on and off the reflectors, or by adding some opacity filter (*e.g.* mirrors that only cast a part of the light beam on the sensor), one can easily measure any weighted sum of light intensity by pixels.
> > > Replacing the optical sensor that measures a value, by a threshold-triggered metal–oxide–semiconductor gives an example for our vector-valued regression setting.
> > >
> > > > I would like to point out that the optimality of the algorithm is only in terms of the dependence on T.
> > >
> > > You are completely right, "minimax optimality" tends to focus only on the number of samples and forget about constants. We were keen in this work to avoid results in $O(T^{-1/2})$ without explicit constants since it is a usual drawback in learning to hide some curse of dimensionality in constants. We were keen to show that our constants do not explode as the output dimension grows.
> > >
> > > > It seems to me that perhaps the "active labeling" general problem lacks enough structure to create effective algorithms, and thus might not be the right abstraction. For example, the algorithms and methods presented require rather general query sets while the practical use cases have strong restrictions on the query sets.
> > >
> > > We understand your concern.
> > > As said above, we try different formalization and the active labeling one seems to us to be the right one (as opposed to *e.g.* formalization were measurements are cast as sigma-algebra, and convergence is studied with tools as in *A Note on the Strong Convergence of $\Sigma$-Algebras* by Hirokichi Kudo (1974)).
> > > It is true that it is quite generic, but we do not think it is too abstract to be hopeless to try to tackle it in a generic fashion - at a similar level of abstraction lies research on structured prediction which has been recognized as useful by the community.
> > > Once again, it is true that the SGD solution we suggest do not easily integrate query constraints, but work-around as in Proposition 4 seem reachable.
> > >
> > > >  I concede that the setting of linear models and one-dimensional outputs (observing thresholds) is reasonable, but this is quite narrow.
> > >
> > > Although we hope that our paper will reach beyond the one-dimensional problem, we do not see this problem as narrow since it has had important applications in pricing (see the work of Maxime Cohen or Renato Paes Leme).
> > >
> > > > I still find the story/scope of the paper scattered without a clear, useful message.
> > >
> > > In our view, the message is "there is this interesting data collection problem that we call "active labeling" and we have a consistent algorithm to tackle it when there is no query restrictions. It leverages the fact that we do not need full information to do SGD".
> > >
> > > Thanks for your implication during this discussion period, and for your comments that help us better understand your concerns.

---

> > > > ### Comment · Reviewer_Evw4 · 2022-08-04
> > > > **Reviewer's response**
> > > >
> > > > The compressed sensing example is a good one.
> > > >
> > > > I like proposition 4 and think this is a good example, though I agree with Reviewer Byvs that Section 7 could use more precision.
> > > >
> > > > I hope that for the next version of the paper, the authors will incorporate some of our discussion above. In particular, fleshing out more examples, connecting sections to a central story, and making sections (especially Section 7) more precise.
> > > >
> > > > Thank you for answering my questions. I still think this paper is borderline, but I will change my score to "borderline accept".

---

> > > > > ### Author Response · Authors · 2022-08-09
> > > > > **Authors' answer**
> > > > >
> > > > > Thank you again for your productive comments, we highly appreciate your inputs.
> > > > > Be sure that we will do our best to integrate them in order to help future readers to get the most of our work.
> > > > > In particular, we will make section 7 more precise, discuss the ranking example, and give more motivations for the vector-valued regression setting.
> > > > > Thinking back of this setting, we thought that beyond compress sensing, pricing products through bundles could be another interesting way to flesh out practical use cases of our algorithm.
> > > > >
> > > > > To be more concrete, $x$ could be some features characterizing a consumer, and $m$ a number of products.
> > > > > Suppose that you want to sell them some baskets of products represented by weight vectors $w \in \mathbb{N}^m$ where $w_i$ represent the number of instances of product $i$ in the basket (similarly, $x$ could be some context for some advertising company and web page, and $m$ could represent the number of advertising spots on this web page; knowing that this web page will be displayed by several anonymous users, you can sell lots with $w_i$ times the advertising spot number $i\in[m]$).
> > > > > Now, assume that $w_i$ could be fractional (*e.g.* you can buy fractional shares in order to replicate the S\&P500 index; or you sell a probability of being displayed at spot $i$), and could also be negative (*e.g.* you are trading derivatives, and can be both short or long).
> > > > > The consumer $x$ is associated with a value $y \in \mathbb{R}^m$ where $y_i$ corresponds to the price they are ready to pay for product $i$.
> > > > > When pricing the basket $w$ at $c$, you observe if the consumer buys it or not, *i.e.* $1_{w^\top y > c}$.
> > > > >
> > > > > Best regards

---

### Official Review · Reviewer_1mW2 · 2022-07-12

**Rating:** 6
**Confidence:** 2
**Soundness:** 4 excellent
**Presentation:** 3 good
**Contribution:** 3 good

**Summary:**

This paper generalizes the standard active learning setup to "active weakly supervised learning," where partial supervision is available instead of full supervision. This paper provides a way to estimate the stochastic gradient when a query of any half-space is given. Statistical analysis and numerical simulation are conducted for simple regression and classification problems to demonstrate the superiority of active strategy over passive strategy.

**Questions:**

- The weak information can give an unbiased gradient estimation. How about the variance?
- Figure 2 (Left) shows that the passive strategy is better than the active strategy for the first 100 steps. Does it mean using a passive strategy is better at the beginning?
- From a practical point of view, the half-space query can be transformed into a binary question, and "Example 1 (Classification with attributes)" can be transformed into multiple-choice questions. What other formats of questions can you think of to facilitate the data annotation process?

Misc
- Line 115: Typo, "we aim at to minimizing" -> "we aim to minimize"
- Line 253: Typo: "ot" -> "to"

**Limitations:**

The authors discuss their limitations in the paper and point out several future directions.

**Strengths And Weaknesses:**

- Originality: This paper is well-motivated and focuses on an interesting problem: how to query the label in a weakly-supervised scenario.
- Quality: The proposed method is novel and supported by extensive theoretical analysis and proof-of-concept numerical simulation.
- Clarity: This paper is well-written with minor flaws.
- Significance: This paper's scope is narrow, mainly focusing on the noiseless median regression problem in a stream setting. But it depicts a potentially new way for the annotator to provide a label, which may lower the bar for the task where an expert is needed.

---

> ### Author Response · Authors · 2022-07-31
> **Answer to Reviewer 1mW2**
>
> Thank you for the time you took to read, understand and review this paper, and for spotting typos.
> We are happy that you have appreciated it.
>
> We would like to point out that our scope is not restricted to the noiseless case, indeed SGD deals really well with noise, which can be seen as an advantage in comparison to zeroth-order methods such as the work of Cohen *et al.* (2020).
>
> We are glad you asked us about the variances of our SGD.
> They are to relate with the inverse of the constants $c_1$ and $c_2$.
> Making sure to have small variance was crucial as convergence rates strongly depend on it: the higher the variance, the slower the convergence.
> We were keen to come up with strategies where those variances do not suffer from the curse of dimensionality (which is a standard drawback for practical applications). They typically scale in $M m^{3/2}$ and not in $M^m$ or $2^m$.
> An exact computation of those constants is given in Appendix B.2.
>
> The difference on Figure 2 between passive and active is due to the fact that the passive and active strategy behave differently with respect to the step size for SGD, we optimize this step size for the last iterate and not for the whole trajectory.
> On Figure 2, the step size seems to be too conservative for the active baseline for the first 100 steps.
>
> Regarding other formats of questions, we did not extend on the ranking example in the paper, we benefit from your last question to do it.
> Ranking is the setting where the output space is the space of permutation over $m$ elements ${\cal Y} = \mathfrak{S}(m)$.
> One way to approach ranking is through the Kendall loss $\ell(y, z) = -\phi(y)^\top \phi(z)$ with $\phi(y) = (1_{y(i) > y(j)})$ for $i < j\leq m$ (the permutation $y$ can be understood as a function from {$1, \cdots, m$} to itself).
> In this setting, the least-square surrogate of Ciliberto *et al.* (2020) consists in learning $g(x) = \mathbb{E}[\phi(Y)\vert X=x]$ as a least-squares problem.
> Hence, our half-spaces translate into the questions $\sum_{i < j\leq m} w(i,j) 1_{y(i) > y(j)} > c$ for some $(w(i,j)), c$ in $\mathbb{R}$. In particular, if we choose $U$ to be uniform on the canonical basis (and not on the sphere), those questions translate into pairwise preferences (*e.g.* does user $x$ prefer movie $i$ or movie $j$?).
> In terms of guarantee akin to Theorem 1, retaking the calibration inequality (Theorem 7) of Ciliberto *et al.* (2020), we get convergence rates of the form $m^{3/2} T^{-1/4}$.
> In terms of guarantee akin to Theorem 2, since we need as least $\log_2(m!) \simeq m\log(m)$ binary queries to discriminate between $m!$ permutations, we can expect a lower bound in $m^{1/2} \log(m)^{1/2} T^{-1/2}$.
>
> We hope to have answers positively to your concerns and stay at your disposal for further questions!

---

> > ### Comment · Reviewer_1mW2 · 2022-08-04
> > **Response to the authors**
> >
> > Thank you for your response.

---

### Meta-Review · Area_Chair_EJfB · 2022-08-26

**Recommendation:** Accept
**Confidence:** Less certain

**Metareview:**

This paper studies “active labeling”, which can be seen as active learning with weak supervision, and proposes an active labeling algorithm based on SGD. The reviewers found that the idea of this paper is innovative. After author response and reviewer discussion, the paper receives generally unanimous support from the reviewers. Thus, I recommend acceptance.

**Award:**

No

---

### Decision · Program_Chairs · 2022-09-14

Accept